# Physiological and Growth Responses of Thai Rice Genotypes to High Salinity Stress at the Seedling and Reproductive Stages

**DOI:** 10.3390/plants14243748

**Published:** 2025-12-09

**Authors:** Supranee Santanoo, Oracha Khianpho, Jirawat Sanitchon, Piyada Theerakulpisut

**Affiliations:** 1Department of Biology, Faculty of Science, Khon Kaen University, Khon Kaen 40002, Thailand; suprsa@kku.ac.th; 2Salt-Tolerant Rice Research Group, Khon Kaen University, Khon Kaen 40002, Thailand; oracha.kp@gmail.com; 3Department of Agronomy, Faculty of Agriculture, Khon Kaen University, Khon Kaen 40002, Thailand; jirawat@kku.ac.th

**Keywords:** local rice varieties, membrane integrity, osmotic adjustment, photosynthesis, salt-tolerance

## Abstract

The objectives of this study were to select Thai rice that are tolerant at the seedling stage and investigate their growth, physiological and yield responses at the reproductive stage in comparison with the standard salt-tolerant Pokkali (PK). Twenty-two local, commercial and improved Thai rice genotypes along with PK and salt-susceptible IR29 were evaluated at the seedling stage for salt tolerance using a 15 dS m^−1^ saline solution with five replications. Ten selected genotypes were grown in pots with four replications and exposed to a 15 dS m^−1^ saline level from early booting to the flowering stage. During the grain-filling stage, salt treatments decreased from 15 to 12 dS m^−1^ and were sustained at this level until harvest. The experimental design for both experiments was a randomized complete block design (RCBD). Based on the physiology of flag leaves, almost all genotypes exceled in the protection of chlorophyll, relative water content (RWC), membrane integrity and lipid peroxidation. In contrast, the photosynthesis, growth and grain yield of all were dramatically reduced. The rice genotypes exhibited varying degrees of osmotic adjustment (OA), ranging from 1.598 to 2.541 MPa. The cultivar RD73 and line TSKC1–144, which were genetically improved from KDML105 by the introgression of a salt-tolerant QTL/gene from PK, showed the least reduction in grain yield (60 and 53% reduction, respectively) along with PK (60%). Among the five Thai local rice varieties, Go Main Surin (GMS) showed the least reduction in grain weight (58%), total plant dry weight (28%) and green leaf dry weight (1%), while Khao Gaew (KG) and Leuang Puang Tawng (LPT) were the most reduced. PK and RD73 showed a high level of tolerance at both the seedling and reproductive stages. In contrast, KG and LPT, which exhibited high tolerance at the seedling stage, showed high susceptibility in growth, yield and most physiological traits. On the other hand, TSKC1–144 was sensitive at the seedling stage but showed increased tolerance at the reproductive stage. This result implies that suitable cultural practices should be performed to obtain the best field conditions for growing rice genotypes having different levels of salt tolerance at the seedling and reproductive stages. Future research should focus on molecular characterization of tolerance mechanisms of the promising local genotypes and the potential to use them as tolerance gene donors.

## 1. Introduction

Rice (*Oryza sativa*) is a vital cereal crop and staple food for over half of the world’s population, particularly in developing countries. Thailand has emerged as one of the world’s top rice exporters, with a cultivation area of approximately 10.82 million hectares [1]. More than half of its rice production originates from the northeast (NE) region. However, only 10% of the rice fields in this region are irrigated and 90% are rainfed lowland paddies, making rice plants more prone to drought and salinity [2]. Saline-affected agricultural land in NE Thailand covers an area of 1.84 million hectares, which is categorized into slightly saline, moderately saline, highly saline and severely saline based on the soil electrical conductivity (EC_e_) of 2–4, 4–8, 8–16 and >16 dS m^−1^, respectively [3]. In NE Thailand, only slightly and moderately saline soils are suitable for rice cultivation, although with significantly reduced crop survival rates and yields [4]. The average production of KDML105, a salt-sensitive elite rice, in moderately saline soils in the NE was as low as 50 g m^−2^ in the low precipitation year of 2018 [5], only 10% of the attainable KDML105 yield of 450 g m^−2^ in non-saline soils [6]. In a highly saline field in NE Thailand, the yield of KDML105 was reduced by 58% compared to that in a nearby non-saline field [7]. This issue limits commercial rice cultivation and adversely affects both the yield and quality of Thai rice.

High salt concentration in saline soils causes low water potential, which leads to osmotic stress in plants, causing water deficit, loss of turgor and reduction in cell expansion and growth, stomatal closure and photosynthesis inhibition [8,9]. In response to salt-induced osmotic stress, plants develop an osmotic adjustment (OA) mechanism by synthesizing and accumulating high concentrations of osmotically active compatible solutes in the cell’s cytosol to reduce the cell’s osmotic potential (OP) and facilitate water uptake from saline soils [10]. Rice genotypes with a greater level of salt tolerance have been reported to display higher OA ability under salt stress associated with more concentrated solutes like sugar and proline [11]. Rice plants growing in saline soil absorb and accumulate Na^+^ to toxic levels, particularly in the leaves, which causes ionic stress, resulting in a disturbance of all major metabolic processes [12]. Compared with salt-sensitive rice, salt-tolerant genotypes possess an efficient Na^+^ exclusion mechanism, accumulate lower toxic Na^+^ in the shoots, and maintain a higher net photosynthetic rate (Pn) under greenhouse [13] and saline field conditions [7]. When subjected to salt stress during the reproductive phase, salt-tolerant genotypes displayed significantly lower Na^+^ and Na^+^/K^+^ in flag leaves and developing panicles, associated with higher pollen viability, percent filled grains and grain weight [14,15]. Salt stress also causes oxidative stress following the overproduction of reactive oxygen species (ROS), which destroy essential macromolecules [16]. Importantly, ROS attack on polyunsaturated fatty acids in the plasma membrane leads to membrane damage, electrolyte leakage (EL) and cell death [17]. Rice genotypes with greater salt tolerance displayed a more remarkable increase in the activity of ROS-scavenging antioxidant enzymes and a lower increase in EL than salt-sensitive rice [18,19]. Therefore, Na^+^ ion exclusion, OA and ROS scavenging are considered the most crucial physiological mechanisms determining salt tolerance in rice.

The response of rice to salinity stress is influenced by multiple factors, including the type and duration of salt exposure, the developmental stage of the plant, and its genetic background [9,12]. Generally, the seedling and reproductive stages are more susceptible to salinity than the germination and vegetative stages [20]. Interestingly, studies have shown that tolerance of some genotypes at the seedling stage is poorly associated with that at the reproductive stage and may be regulated by different sets of genes or quantitative trait loci (QTLs) [21,22]. Research on mapping QTLs for salt tolerance in rice at the seedling stage has advanced significantly in the last two decades, resulting in the identification of several hundred QTLs from different genetic backgrounds of biparental mapping populations [21,23]. Although a higher correlation has been found between yield and salt tolerance at the reproductive stage than at the seedling stage, only a few QTL studies have been carried out at the reproductive stage, and a limited number of QTLs have been identified [24]. The detection of *Saltol* QTL, a major QTL controlling shoot K^+^/Na^+^ homeostasis at the seedling stage from a recombinant inbred line (RIL) derived from a cross between IR29 (salt-sensitive) and Pokkali (salt-tolerant landrace) [25], was an important landmark in salinity tolerance breeding. The *Saltol* QTL has been most widely used in molecular breeding for integration into elite salt-sensitive rice in several countries to improve salt tolerance [21,26]. The *Saltol* QTL contains the gene *SKC1* encoding HKT1;5 transporter protein, which is primarily responsible for Na^+^ exclusion from shoots, resulting in the improvement of salt tolerance in elite rice [27]. However, some lines of improved rice introgressed with *Saltol* were more tolerant than the sensitive parent only at the seedling stage but still sensitive to salinity and produced low yield at the reproductive stage [21,28]. Therefore, more studies are needed on physiological tolerance mechanisms and the identification of reproductive stage QTLs so that QTLs at both developmental stages can be pyramided to obtain superior improved rice cultivars.

Most of the molecular breeding programs globally have employed *Saltol* QTL, from a limited number of salt-tolerant landrace donors like Pokkali or Nona Bokra, to increase salt tolerance of the sensitive elite rice primarily through Na^+^ exclusion from shoots [21]. Nevertheless, salt tolerance in rice is also governed by additional mechanisms, including OA, tissue tolerance, antioxidant capacity and morpho-anatomical adaptation [29]. These traits may not be well expressed in traditional donors such as Pokkali and Nona Bokra, highlighting the need to explore novel genes/QTLs responsible for those mechanisms in wild and local rice germplasms for use in pyramiding multi-tolerance mechanisms into elite rice [26,30]. Rahman et al. [31] identified 7, out of 82, Bangladeshi landraces that are salt-tolerant but have different alleles from that of Pokkali at the *Saltol* locus. In addition to having low shoot Na^+^, some of these landraces have significantly higher relative water content (RWC) and lower membrane damage than Pokkali, suggesting that they are potential novel donors with new tolerance mechanisms. Thailand has significant potential for discovering salt-tolerant traits within its indigenous rice varieties, with over 17,000 local accessions conserved at the National Rice Gene Bank [32]. Therefore, to broaden the genetic diversity of salt-tolerant donors, this study was designed to select indigenous local Thai rice that are salt-tolerant at the seedling stage and to investigate their physiological responses and yield at the reproductive stage. Underutilized salt-tolerant local rice with good tolerance at both developmental stages may be employed as tolerant gene donors in breeding programs and recommended for cultivation in saline areas. For comparison of physiological mechanisms, the salt-sensitive elite Thai rice KDML105 together with RD73 and TSKC1–144, which are genetically improved from KDML105 through introgression of *Saltol* QTL and *SKC1*, respectively, will be included in this study. Information regarding the differential levels of tolerance at the seedling and reproductive stages obtained from this study will provide an important implication for selecting genotypes for breeding and for management of rice productivity in saline fields.

## 2. Results

### 2.1. Screening for Salt Tolerance at the Seedling Growth Stage

To evaluate the salt tolerance of 24 rice genotypes at the seedling stage, the salt treatments were started at 14 days after germination (DAG) for 10 days by immersing in a 15 dS m^−1^ saline solution (150 mM NaCl), while tap water was given to the control sets. Morphological responses under non-saline and saline conditions are shown in Figure 1. Salt-injury assessments revealed distinct differences in seedling tolerance among the rice genotypes, with several local varieties performing as well as, or better than, the tolerant check Pokkali. In contrast, IR29 consistently exhibited the highest sensitivity to salinity. Salt-injury scores were recorded at 7 and 10 days after salt exposure following the IRRI scale (1 = highly tolerant, 3 = tolerant, 5 = moderately tolerant, 7 = susceptible, 9 = highly susceptible; Figure 2). After 7 days, Khao Gaew, Leuang Tah Yang, Pahn Tawng60, PTT1, LLR395, RD73, Hom Dang Nouy, and Hom Noun (scores 3.0–3.6) showed significantly lower salt-injury scores (*p* < 0.01) compared with IR29, Leuang Puang Tawng, TSKC1–144, Luang Pratahn, Ma Hom, HMLN, Lou Tang and Mali, which ranged from 5.2 to 7.0 (Figure 2A; Appendix A). At 10 days, Pahn Tawng60, Go Main Surin, Hom Noun, LLR395, Puang Tawng, RD73, Khao Gaew and Leuang Puang Tawng did not differ significantly from Pokkali. Notably, the local variety Hom Dang Nouy exhibited an even lower score than Pokkali (Figure 2B), indicating superior tolerance. IR29 remained the most sensitive genotype (score = 9), while all remaining genotypes scored between 5.0 and 7.4 (Figure 2B).

Salt stress had a significant effect on leaf greenness, plant height and total biomass of the rice genotypes (*p* < 0.01, Appendix A). Under control conditions, leaf SPAD values were not significantly different among genotypes (*p* > 0.01), with a mean SPAD of 31.48 across all genotypes (Figure 3A). In contrast, seedlings grown under saline conditions showed a significant reduction, to 23.24, in mean SPAD values, representing a 26% decrease relative to the control. SPAD values of 12 genotypes of salt-treated plants, including Pokkali, Leuang Puang Tawng, LLR395, Hom Dang Nouy, Hom Noun, Puang Tawng, Pahn Tawng59, Khao Gaew, RD73, Go Main Surin, Pahn Tawng60, Khao Supan, Mali, and Ma Hom (ranging from 21.30 to 32.84), displayed non-significant reduction from those of the control plants. On the other hand, SPAD values of the genotypes RD61 (19.98), HMLN (19.60), PTT1 (19.02), Gon Gaew (18.22), KDML105 (18.18), TSKC1–144 (16.76), Leuang Tah Yang (15.98), Luang Pratahn (15.56), Lou Tang (14.68) and IR29 (7.52) significantly (*p* < 0.01) reduced under salinity (Figure 3B).

Plant height and total biomass were measured at 24 DAG (Figure 3C–F). The mean plant height under control conditions was 62.57 cm, whereas that of the stressed plants was significantly lower (36.91 cm; 40% reduction; *p* < 0.01). Under control conditions, plant heights of Khao Supan, PTT1, Hom Noun, Pahn Tawng60, Pokkali and Khao Gaew (ranging from 69.00 to 74.40 cm) were significantly higher than those of Leuang Puang Tawng, Gon Gaew, Ma Hom, Puang Tawng, LLR395, RD61, IR29 and HMLN (ranging from 44.60 to 62.20 cm), whereas the remaining genotypes showed intermediate heights (Figure 3C). Under salinity, PTT1, Khao Supan, Leuang Tah Yang, Hom Noun and Pokkali (ranging from 40.60 to 50.40 cm) exhibited significantly greater height (*p* < 0.01) than Mali, RD73, Ma Hom, LLR395, RD61, Puang Tawng, HMLN, and IR29 (ranging from 27.10 to 34.90 cm) (Figure 3D).

For total dry biomass, the mean value across control plants was 0.36 g plant^−1^, significantly higher (*p* < 0.01) than that of saline plants (0.14 g plant^−1^), representing a 59% reduction. Under control conditions, the highest biomass was observed in Pahn Tawng60, Hom Noun, Pokkali, Khao Supan, Pahn Tawng59 and Leuang Tah Yang (ranging from 0.48 to 0.42 g plant^−1^). In contrast, Ma Hom, Mali, LLR395, RD61, Lou Tang, PTT1, TSKC1–144, HMLN and IR29 showed the lowest biomass (ranging from 0.18 to 0.33 g plant^−1^; Figure 3E). Under salinity, Pokkali produced the highest biomass (0.21 g plant^−1^), significantly higher than Hom Dang Nouy, Gon Gaew, Ma Hom, Lou Tang, Leuang Tah Yang, KDML105, TSKC1–144, PTT1, Mali, HMLN and IR29, whose biomass ranged from 0.10 to 0.13 g plant^−1^ (Figure 3F). Other genotypes showed intermediate values.

For a clear visualization and classification of all rice genotypes, a heatmap combined with an agglomerative hierarchical clustering analysis (HCA) was made, and the results are illustrated in Figure 4. Based on the mean values of salt-injury score, SPAD, plant height and dry total biomass, the HCA clearly divided the rice plants into two major clusters (“A” and “B”). Cluster A included rice genotypes grown under non-saline conditions, while cluster B included rice plants grown under saline conditions. For cluster B, 24 rice genotypes were divided into two minor groups (“a” and “b”); the standard rice check Pokkali was in group “a”, while the susceptible rice check IR29 was in group “b”. The high tolerance (HT) and tolerance (T) rice genotypes were clustered in group “Ba1,” including Pokkali, Khao Supan, Leuang Puang Tawng, Hom Noun, Pahn Tawng59, Pahn Tawng60, Khao Gaew, Puang Tawng, Go Main Surin, LLR395, RD73 and Hom Dang Nouy. Rice genotypes that are moderately tolerant (MT), moderately susceptible (MS) and susceptible (S) were grouped together in “Bb3,” including Gon Gaew, HMLN, TSKC1–144, Lou Tang, Luang Pratahn, Leuang Tah Yang, Ma Hom, Mali, PTT1, RD61 and KDML105. The highly susceptible (HS) IR29 was the only genotype classified in the “Bb2” group (Figure 4).

### 2.2. Physiological Responses and Growth of Rice at the Reproductive Stage

To investigate rice physiological responses and growth under salinity stress at the reproductive stage, ten rice genotypes, including six local Thai rice varieties—namely Leuang Puang Tawng (LPT), Hom Noun (HN), Khao Gaew (KG), Go Main Surin (GMS), LLR395, and Hom Dang Nouy (HDN)—two improved salt-tolerant rice lines (RD73 and TSKC1–144), one susceptible genotype (KDML105), and the standard salt-tolerant rice (Pokkali) were selected from the screening of salt tolerance at the seedling stage of 24 rice genotypes. After 63 days of germination (Appendix A), salt treatments were initiated at the early booting stage by irrigating with 150 mM NaCl for 19 days, after which the concentration was reduced to 120 mM NaCl and maintained until harvest. Morphology and growth at the reproductive stage of the 10 rice genotypes growing under non-saline (C) and saline conditions (S) are displayed in Figure 5.

#### 2.2.1. SPAD and Chlorophyll Contents

The mean SPAD values and chlorophyll content under salt stress were non-significantly different from those of the controls; however, SPAD values showed significant differences (*p* < 0.01) among rice genotypes. The SPAD value of the control plants of LLR395 (42.25) was significantly higher than other genotypes, followed by PK (37.97), while KG (35.45) was significantly lower than LLR395 and PK (Figure 6A). For the SPAD values of the stress plants, LLR395, PK and HDN (ranging from 39.83–38.68) showed significantly higher values than HN, KDML105 and TSKC1–144 (ranging from 34.23–35.63), while other genotypes showed intermediate values. KG under salt conditions showed a significantly higher SPAD value than the control (+7%), while that of the stressed GMS had a significantly lower value than the control (approximately 4% reduction). Total chl, Chl a, Chl b and Chl a/b were non-significantly different among rice genotypes under both control and stressed conditions (Figure 6B–E). The control plants of LLR395 and HDN showed the highest values of total chl (3.36 and 3.35 mg g^−1^ FM, respectively), Chl b (1.97 and 1.86 mg g^−1^ FM) and Chl a/b (1.97 and 1.86), while RD73 and HDN showed the highest Chl a (1.48 mg g^−1^ FM). Under salt stress, KG displayed the highest total chl (3.39 mg g^−1^ FM), Chl b (2.04 mg g^−1^ FM) and Chl a/b (2.04 mg g^−1^ FM), while HDN had the highest Chl a (1.46 mg g^−1^ FM).

#### 2.2.2. Photosynthetic Performance

The patterns of the light response curve of the effective quantum yield of PSII photochemistry (Φ PSII) and electron transport rate (ETR) under control and salt stress conditions showed differential responses to varying light intensity (Figure 7A−D). The mean Φ PSII and ETR under PAR 1200 μmol (photon) m^−2^ s^−1^ across genotypes of the salt stress plants (0.162 and 82.42 µmol e^−1^ m^−2^ s^−1^, respectively) showed significantly (*p* < 0.01) reduced values from the control (0.194 and 102.28 µmol e^−1^ m^−2^ s^−1^) (Figure 7E,F). The Φ PSII and ETR of LLR395, HDN, LPT and KG under salt stress were significantly decreased compared with the controls (approximately −16%, −27%, 41% and −43%, respectively), while the values of other rice genotypes were not significantly different from the controls (Appendix A).

The photosynthetic light-response curves (Pn/I curve) of ten rice genotypes growing under control and salt-stress conditions are depicted in Figure 8. Under control conditions, the Pn of most genotypes approached the saturation point at the PAR of 600 μmol (photon) m^−2^ s^−1^, except for GMS, HDN, KG and TSKC1–144, whose Pn tended to continue increasing further with increasing light intensity. Under salinity, the Pn of all rice genotypes was inhibited at all levels of light intensity, with some exceptions. The variety KG subjected to salt stress, when illuminated with low light intensity of 50 and 200 μmol (photon) m^−2^ s^−1^, exhibited higher Pn compared with the control, indicating shade adaptation (Figure 8D). Under salinity stress, the PAR levels at which Pn approached saturation varied among genotypes: 200 μmol (photon) m^−2^ s^−1^ for KG; 600 μmol (photon) m^−2^ s^−1^ for GMS, HDN, HN and LPT; and above 600 μmol (photon) m^−2^ s^−1^ for LLR395, KDML105, RD73, TSKC1–144 and PK.

Photosynthetic performance under a PAR of 1200 μmol photons m^−2^ s^−1^ showed substantial reductions in carbon assimilation, stomatal conductance, and transpiration under salinity, while water-use efficiencies (WUEi and WUE) increased slightly under stress. The mean Pn across genotypes of the control plants (12.91 µmol CO_2_ m^−2^ s^−1^) was significantly (*p* < 0.01) higher than that of the stressed plants (4.77 µmol CO_2_ m^−2^ s^−1^). Under control conditions, the Pn of LLR395 and RD73 (17.54 and 14.37 µmol CO_2_ m^−2^ s^−1^, respectively) was significantly higher (*p* < 0.05) than that of KG and HDN (11.07 and 10.04 µmol CO_2_ m^−2^ s^−1^, respectively). Under salt stress, all rice genotypes exhibited significant reductions in Pn, varying from 48 to 81% (Appendix A). However, RD73 showed a significantly higher (*p* < 0.05) Pn (6.80 µmol CO_2_ m^−2^ s^−1^) compared to PK, HN, LLR395 and HDN, whose values ranged from 3.21 to 3.51 µmol CO_2_ m^−2^ s^−1^ (Figure 9A and Appendix A). The mean gs across genotypes of control plants was 0.169 mol H_2_O m^−2^ s^−1^, while that of salt stress plants was 0.048 mol H_2_O m^−2^ s^−1^ (67% reduction). Under non-saline conditions, gs of KDML105, LLR395, RD73, GMS and HDN (ranging from 0.143 to 0.246 mol H_2_O m^−2^ s^−1^) were significantly higher than HDN, LPT and KG (ranging from 0.098 to 0.109 mol H_2_O m^−2^ s^−1^) (Figure 9B). Under salt-stress, gs of all rice genotypes, except for KG, significantly (*p* < 0.01) decreased from the control, with the values ranging from 0.026 to 0.062 mol H_2_O m^−2^ s^−1^). As a result of stomatal closure, the mean transpiration rate (Tr) across genotypes of the stressed plants (1.629 mmol H_2_O m^−2^ s^−1^) was significantly (*p* < 0.01) reduced (−64%) from that under the non-saline condition (4.751 mmol H_2_O m^−2^ s^−1^) (Figure 9C). For the control plants, the Tr of LLR395 and GMS (5.901 and 5.479 mmol H_2_O m^−2^ s^−1^, respectively) was significantly higher (*p* < 0.01) than that of KG, PK, and LPT (ranging from 4.145 to 4.201 mmol H_2_O m^−2^ s^−1^), while other genotypes had intermediate Tr values. When subjected to salt stress, LPT showed the lowest Tr (0.720 mmol H_2_O m^−2^ s^−1^), while those of RD73 and KDLM105 were the greatest (1.991 and 1.976 mmol H_2_O m^−2^ s^−1^, respectively).

Although the mean WUEi and WUE across genotypes were slightly higher in stressed rice plants compared to the control plants, the differences were not statistically significant. For the control plants, LPT had the highest WUEi (126.62 µmol CO_2_ mol H_2_O^−1^) and a significantly higher WUEi (*p* < 0.01) than that of HDN, GMS, RD73 and KDML105 (ranging from 51.97 to 77.59 µmol CO_2_ mol H_2_O^−1^) (Figure 9D,E). Under salt-stress, LPT also had significantly (*p* < 0.01) the highest WUEi (181.06 µmol CO_2_ mol H_2_O^−1^), followed by KG (138.64 µmol CO_2_ mol H_2_O^−1^). The lowest WUEi was observed in HN and PK (64.62 and 66.76 µmol CO_2_ mol H_2_O^−1^, respectively). The significantly increased WUEi values under salt stress were observed in only KDML105 (+149%) and LPT (+41%) (Appendix A). The stressed LPT plants also showed the highest WUE (6.53 µmol CO_2_ mmol H_2_O^−1^), which was considerably higher (*p* < 0.01) than all other genotypes (Figure 9E). The WUE of PK, HN and HDN (1.9 µmol CO_2_ mmol H_2_O^−1^) were the lowest under salinity. A significant increase in WUE under salt stress was observed in only LPT (+97%), while the other genotypes showed non-significant increases (Appendix A). The mean Ci/Ca across genotypes under salt stress (0.514) was slightly lower than that under control conditions (0.602), but not statistically different. Under control conditions, the Ci/Ca of KG and LPT (0.4) was the lowest, while KDML105, RD73 and GMS showed a significantly higher Ci/Ca (ranging from 0.675 to 0.748) (Figure 9F). Under salt stress, KG, KDML105 and LPT displayed a significantly (*p* < 0.01) lower Ci/Ca (ranging from 0.241 to 0.398) than GMS, HDN, PK and HN (ranging from 0.633–0.694). A significant decrease in Ci/Ca under salinity was observed in only RD73 (−32%) and KDML105 (−49%) (Appendix A).

#### 2.2.3. Relative Water Content (RWC), Osmotic Potential (OP) and Osmotic Adjustment (OA)

Although the mean RWC across genotypes was slightly lower in stressed rice plants (88.67%) compared to the control (92.21%), the difference was not statistically significant (Figure 10A, Appendix A). Among the ten rice genotypes, a significant decrease in RWC under salinity stress was observed only in rice variety LPT (−6%), KG (−7%), and RD73 (−9%), while no significant changes were detected in the other genotypes. Rice genotypes under non-saline conditions displayed a similar OP (−1.278 to −1.416 MPa). Under salt stress, the OP values of all ten genotypes significantly reduced compared with the controls (Figure 10B, Appendix A) and were significantly different among genotypes, indicating active accumulation of solutes in response to stress. The stressed plants of KG displayed the lowest OP (−3.149 MPa), followed by LPT (−2.945 MPa) and GMS (−2.757 MPa), for which the OP values reduced by 143, 122 and 107% compared with the controls. On the other hand, the OP of PK was the highest (−1.886 MPa) and the least reduced (46% reduction). Rice plants actively accumulated inorganic and organic solutes to osmotically adjust the cell’s water status to the low water potential outside. Rice genotypes varied in the osmotic adjustment (OA) activity, showing the highest OA in KG and LPT (2.576 and 2.541 MPa, respectively) and the lowest in PK (1.598 MPa), with the mean across genotypes of 2.12 MPa (Figure 10C, Appendix A). The ten genotypes were clearly divided into two groups. The high OA group included KG, LPT, LLR395, GMS and KDML105, with OA ranging from 2.576 to 2.283 MPa, and the low OA group (PK, RD73, HDN, HN and TSKC1–144, with OA varying from 1.598 to 1.873 MPa).

#### 2.2.4. Electrolyte Leakage (EL), Malondialdehyde (MDA) and Ion Concentration

The degree of membrane injury and the level of oxidative damage from salt stress were indicated by EL and MDA values, respectively. Salt stress damaged cellular membranes of all genotypes, except RD73, resulting in a tendency for EL to increase from 9.14 to 15.13%, compared with 7.19 to 11.61% in non-saline plants (Figure 11A, Appendix A). However, only HN displayed a significantly (*p* < 0.05) higher EL than the control (78% increase from 8.55 to 15.11%). The degree of ROS attack on lipids is indicated by the MDA content, one of the final products of lipid peroxidation. The MDA content of most genotypes remained stable under salt stress, except for HDN, RD73, TSKC1–144 and HN, which showed a relatively high increase, ranging from 25 to 40% (Figure 11B, Appendix A). However, only HN showed a significant increase in MDA (+40%) and had the highest MDA content (Appendix A). The significant increase in the MDA of HN was related to the significant increase in EL. This indicated that the flag leaves of most rice genotypes were quite resistant to oxidative damage. 

The Na^+^ content and the Na^+^/K^+^ ratio in the shoot tissues of salt-stressed plants are considered the most reliable indicators for salt tolerance in rice; the lower they are, the more tolerant the plants. Under salt stress, all rice genotypes exhibited a massive increase in Na^+^ (19 to 80 times, Figure 11A and Appendix A). Among local Thai rice, HDN exhibited the lowest Na^+^ (1.68%), while LPT showed the highest (2.72%) (Figure 11C and Appendix A). As expected, PK accumulated the lowest Na^+^ (1.09%) due to its efficient ion exclusion mechanism. The cultivar RD73 and line TSKC1–144, which were improved from the KDML105 genetic background through an introgression of a gene for sodium exclusion (*SKC1*) from PK, had significantly lower Na^+^ than its sensitive parent KDML105. The K^+^ concentration in stressed plants mostly reduced or remained stable except for LLR395, LPT and PK, which exhibited a significant increase (Figure 11D). Following the increase in Na^+^, the Na^+^/K^+^ of the flag leaves of all ten genotypes steeply increased from an average of 0.03 in the control plants to 1.39 (Figure 11D, Appendix A). As expected, the standard check PK exhibited the lowest Na^+^/K^+^ (0.491), while the sensitive KDML105 had the second-highest Na^+^/K^+^ (1.836). The improved rice RD73 and TSKC1–144 had significantly lower Na^+^/K^+^ than the sensitive parent KDML105. Among local Thai rice, HDN had the lowest Na^+^/K^+^ due to its lower Na^+^ compared to the others. Although LLR395 accumulated the highest Na^+^, its high efficiency in absorbing K^+^ led to its ability to maintain good ion homeostasis with low Na^+^/K^+^.

#### 2.2.5. Total Sugar and Starch

Accumulation of sugar and starch in flag leaves reflects the balance between CO_2_ assimilation and photosynthate export to developing grains. Under salt stress, rice plants exhibited higher sugar content across genotypes (20.33 mg g^−1^ FW) compared to control plants (15.86 mg g^−1^ FW), although the difference between treatments was not statistically significant (Figure 12A and Appendix A). Under non-saline conditions, TSKC1–144 (24.25 mg g^−1^ FW) and GMS (23.99 mg g^−1^ FW) showed the highest total sugar, followed by PK, LLR395 and HDN (18.77, 17.05, and 16.14 mg g^−1^ FW, respectively). In contrast, HN (9.65 mg g^−1^ FW) and KG (9.53 mg g^−1^ FW) exhibited the lowest values. Under salt stress conditions, the total sugar of KDML105 (27.35 mg g^−1^ FW) was the highest, while that of GMS, KG and LPT was the lowest. The significant increases in total sugar under salt stress, compared to the control conditions, were observed in HN (+72%), KDML105 (+85%) and KG (+143%), which may indicate a reduction in photosynthate export from flag leaves and the role of sugar in OA.

The mean starch content across genotypes between the control (6.14 mg g^−1^ FW) and salt conditions (8.73 mg g^−1^ FW) was not significantly different (Figure 12B, Appendix A). All genotypes, except KG, tended to store starch in response to salt stress, particularly HN and LPT, which showed a significant increment of 344 and 198%, respectively. Rice genotypes accumulated different forms of carbohydrates in response to salt stress. KDML105 and KG accumulated photosynthate preferentially in the form of sugar, while LPT preferentially stored starch. On the other hand, HN significantly accumulated both sugar and starch.

#### 2.2.6. Growth, Biomass and Grain Weight

Salinity caused significant (*p* < 0.05 and *p* < 0.01) reductions in plant height, stem DW, root DW, grain DW, and total DW, but a significant increase in DW of senescent leaves (SL); however, several genotypes exhibited partial resilience in particular growth traits (Figure 13 and Appendix A). The mean plant height at the final harvest (110 DAG) across genotypes of the stressed plants was 121.98 cm, while that of the control plants was 143.12 cm. Among rice genotypes, PK showed the highest plant height (180.3 cm for control and 149.5 cm for salt stress), while LLR395 had the lowest plant height (104.3 cm for control and 83.3 cm for salt stress) (Figure 13A). All rice genotypes under salt stress displayed significantly decreasing plant height relative to the control (approximately 9–24%), except RD73 (−6%) and TSKC1–144 (−2%), which showed non-significant reduction (Appendix A).

The mean total leaf DW across genotypes between the control and salt-stress conditions was not significantly different. Rice cv. KG exhibited the highest total leaf DW under both control (7.27 g plant^−1^) and salt-stress conditions (6.60 g plant^−1^), whereas the HN genotype showed the lowest values (3.22 and 3.27 g plant^−1^, respectively) (Figure 13B). Interestingly, GMS and PK under salt stress produced higher total leaf DW than the control (Figure 13B). The green leaf DW of stressed plants of PK (4.09 g plant^−1^) and KG (4.08 g plant^−1^) was significantly higher than that of KDML105, RD73, TSKC1–144, HN and LLR395 (ranging from 1.39 to 1.86 g plant^−1^) (Figure 13C). Significant decreases in GL DW under salinity stress were observed in KG (−31%) and LPT (−38%) (Appendix A). The senescent leaf DW was not significantly different among the rice cultivars in each treatment (Figure 13D). However, the senescent leaf DW of LLR395 and GMS (3.44 and 3.15 g plant^−1^, respectively) was higher than that of other genotypes under salinity. The highest stem DW of control plants was found in KG (34.67 g plant^−1^) and LPT (28.16 g plant^−1^) while PK (15.92 g plant^−1^) and KG (10.25 g plant^−1^) displayed the highest stem DW under salt stress (Figure 13E). Significant reductions in stem DW under salt stress were observed in the genotypes HN, PK, TSKC1–144, LPT, and KG, with the reductions of 36%, 50%, 54%, 66%, and 69%, respectively (Appendix A). Among the rice genotypes, KG had the highest root DW under both control (5.81 g plant^−1^) and salt stress conditions (2.34 g plant^−1^) (Figure 13F). In contrast, HDN and KDML105 showed significantly lower root DW under control conditions (1.84 and 1.71 g plant^−1^, respectively) compared to KG. Under salt stress, no significant differences in root DW were observed among the genotypes. However, significant reductions in root DW from control were detected only in KG and LPT, which showed −47% and −44% reduction, respectively (Appendix A). The mean DW of grains across genotypes under salt stress was significantly (*p* < 0.01) reduced from 9.86 g plant^−1^ in the control condition to 2.13 g plant^−1^ (78% reduction) (Appendix A). Under the non-saline condition, grain weight per plant was significantly different among genotypes ranging from 7.01 g plant^−1^ (KG) to 13.68 g plant^−1^ (LLR395). Under salt stress, the significant difference between genotypes was observed only between the most productive GMS (4.15 g plant^−1^) and the least productive KG (0.46 g plant^−1^) while the remaining genotypes produced lower grain weight (0.73 to 3.27 g plant^−1^) but not significantly different from GMS (Figure 13G; Appendix A). For total biomass, KG (54.78 g plant^−1^) and LPT (45.55 g plant^−1^) showed the highest total dry weight (DW) under non-saline conditions, whereas PK and GMS recorded the highest total DW under salinity stress (Figure 13H). Significant reductions in total dry weight (DW) were observed in all rice genotypes except the salt-tolerant standard, PK (Appendix A).

#### 2.2.7. Principal Component Analysis (PCA) and Hierarchical Clustering Analysis (HCA)

To clearly visualize the relationships among ten rice genotypes under different salt treatments, principal component analysis (PCA) and a heatmap based on agglomerative hierarchical clustering analysis (HCA) were performed (Figure 14). The PCA of 31 parameters, including photosynthetic pigments (SPAD, Chl a, Chl b and Chl a/b), chlorophyll fluorescence (Φ PSII and ETR), leaf gas exchange at a light intensity of 1200 µmol m^−2^ s^−1^ (Pn, gs, Tr, WUEi, WUE, Ci and Ci/Ca), RWC, OP, OA, EL, Na^+^, K^+^, Na^+^/K^+^ ratio and growth (plant height, DW of total leaves, green leaves, stem, root and grain) were generated to determine the parameters that were the major contributors to the variations. PC1 and PC2 explained 39.84% and 17.29% of the overall variation, respectively (Figure 14A). Photosynthetic performance (ETR, Pn, gs and Tr), OP, grain weight, RWC, and total DW were positively related to PC1, whereas Na^+^, Na^+^/K^+^ ratio, OA, senescent leaf DW and starch were negatively related to PC1. The DW of total leaf, green leaf, root, stem, and WUEi were positively related to PC2. On the other hand, Ci/Ca and K^+^ were negatively related to PC2. The PCA and HCA analyses of physiological and growth responses clearly separated the rice genotypes into two major clusters, i.e., “G1” and “G2,” which included rice genotypes under control and salt stress conditions, respectively. Significant parameters that separated the two groups included Na^+^, Na^+^/K^+^ ratio, OA and DW of senescent leaves, which consistently increased in all genotypes under salt stress, while leaf gas exchange parameters (Pn, gs, and Tr), OP, and grain weight consistently decreased (Figure 14B). Rice varieties KG and LPT under control conditions and salt stress conditions were grouped in cluster “G1a” and “G2a”, respectively. The remaining eight genotypes (GMS, HDN, HN, LLR395, KDML105, PK, RD73 and TSKC1–144) under control conditions were clustered in “G1b” and those under salt stress in “G2b” (Figure 14B). This indicated that the physiological and growth characteristics of KG and LPT under both conditions were similar and clearly differed from the others.

## 3. Discussion

### 3.1. Salt Tolerance at the Seedling Stage

After salt stress treatment of seedlings for 10 days, the 24 rice genotypes exhibited variable levels of salt injury and growth inhibition (Figure 2 and Figure 3; Appendix A). The standard salt-tolerant variety Pokkali (PK) showed a high level of tolerance, with an SES score of 3.4, while all seedlings of the standard salt-sensitive IR29 were dead and displayed the highest score of 9.0. While plant height and total biomass of all 24 genotypes were significantly reduced, a significant reduction (−38 to −76%) in leaf greenness or SPAD values was observed in only 10 genotypes (Appendix A). The SPAD values of 13 genotypes non-significantly reduced (−1 to −29%), and that of PK even slightly increased (+7%). This result demonstrated that salt stress had more damaging effects on growth inhibition than chlorophyll degradation [18,33]. In addition to salt injury symptoms, SPAD values may be employed as a reliable indicator for screening of salt tolerance of rice at the seedling stage due to the ease of measurement and varietal differences in response to salt stress.

Among the 16 local Thai rice varieties, all exhibited lower SES scores compared to the standard salt-sensitive cultivar IR29 (Figure 2B). Additionally, 11 varieties (GMS, HDN, HN, KG, KS, LPT, MH, Mali, PT, PT59, and PT60) maintained higher SPAD values than did IR29, indicating better leaf greenness under high salinity conditions (Figure 3B). Based on SES scores at 10 days, HDN at the seedling stage was even slightly more tolerant than PK (Figure 2B and Appendix A). According to the clustering analysis (Figure 4), 9 out of 16 Thai local rice genotypes were tolerant or moderately tolerant and clustered in the same group as PK, i.e., ‘Ba1’ (the tolerant and moderately tolerant group), while the other 7 genotypes were in ‘Bb3’ (the moderately sensitive and sensitive group). For the most widely grown commercial rice for Thailand, KDML105 and PTT1, both were salt-sensitive, having an SES score of 6.6, similar to previous reports [34,35]. This emphasizes the need for the improvement of elite rice cultivars. The cultivar RD73, which was genetically improved from KDML105 by the introgression of *Saltol* QLT, was much more tolerant than KDML105, based on significantly lower SES scores and SPAD values (Figure 2A and Figure 3B; Appendix A), indicating the significant salt tolerance functions of genes located on the QTL [28]. In contrast, the improved line TSKC1–144 from KDML105, introgressed with the *SKC1* gene, was slightly more sensitive than KDML105. Seedlings of this line showed higher tolerance than KDML105 when grown in hydroponic solution treated with 150 mM NaCl [36] but were less tolerant when grown in the soil system in this study. This could be due to the small soil volume, which limited root proliferation in the current study.

For successful production of rice genotypes in saline soils, they need to be salt-tolerant at both the seedling and reproductive stages. Therefore, the five tolerant to moderately tolerant local rice genotypes (HDN, LPT, KG, HN and GMS) and one KKU germplasm (LLR395) in the ‘Ba1’ group (Figure 4) with SES scores ranging from 3.2 to 4.6 (Figure 2B; Appendix A) were selected for an evaluation of growth, yield and physiological responses to high salt stress at the reproductive stage, in comparison with PK, KDML105, RD73 and TSKC1–144.

### 3.2. Physiological Responses, Growth and Yield in Response Salt Stress at the Reproductive Stage

The ten rice genotypes responded differently at the reproductive stage in many aspects of growth and physiology. The impact of salinity on chlorophyll content in rice has been reported to cause a decline in both chlorophyll a and chlorophyll b, due to high salt levels interfering with chloroplast structure and function [37]. However, in this study, the chlorophyll contents of flag leaves of all ten genotypes were not affected by salt stress and showed no significant differences among rice genotypes (Figure 6; Appendix A). This indicated that flag leaves of all ten genotypes were tolerant to chlorophyll degradation at the reproductive stage, including KDML105 and TSKC1–144, which displayed significant reduction in leaf greenness at the seedling stage (Appendix A). The results in this study were consistent with a previous experiment at the booting stage, where Hom Chan, a tolerant indigenous Thai rice, showed no reduction in chlorophyll content, while that of the highly sensitive elite rice cultivar, PT1, was reduced by 45% [38]. In contrast to chlorophyll content, salt stress drastically inhibited photosynthesis in all genotypes, resulting in a 48 to 81% reduction in net photosynthesis rate (Pn) (Figure 9A, Appendix A). This was similar to a recent report on five genotypes of Thai local rice subjected to salt stress at the booting stage [39]. The reduction in Pn was primarily associated with limited CO_2_ uptake due to stomatal closure, as evidenced by a 55–88% reduction in stomatal conductance (gs) (Figure 9B, Appendix A). Under salinity stress, rice plants close their stomata to reduce transpiration (Figure 9C) and conserve water, but this response also limits CO_2_ uptake, leading to a reduction in CO_2_ assimilation [7]. However, the transpiration rate (Tr) and gs decreased more drastically than Pn, leading to an increase in WUE and WUEi in most genotypes (Figure 9D,E). The outstanding WUE of LPT could be associated with its efficient stomatal regulation, attributed to greater reductions in stomatal density [40] and more rapid stomatal closing [41]. The reduction in Pn in some genotypes (HDN, LLR395, KG and LPT) could also be attributed to a significant reduction in PSII photochemical efficiency followed by reduced electron transport (Figure 7D,E), which limited the supply of NADPH and ATP for Calvin cycle operation and induced ROS accumulation [42]. KDML105 and RD73 under stress displayed a significant reduction in the Ci/Ca ratio compared to the control, at approximately −49% for KDML105 and −32% for RD73 (Figure 9F; Appendix A), indicating that photosynthesis in these genotypes was primarily limited by stomatal closure, leading to a relatively high Pn under stress (6.07–6.80 µmol CO_2_ m^−2^ s^−1^; Appendix A). In contrast, the other rice genotypes showed a stable or increased Ci/Ca ratio, indicating that Pn was limited by both stomatal and non-stomatal limitations under salinity stress, leading to relatively low Pn (3.20–5.44 µmol CO_2_ m^−2^ s^−1^; Appendix A) [43]. Reduced Pn under salinity via non-stomatal limitation was ascribed to the reduction in the rates of RuBP carboxylation, RuBP regeneration and triose phosphate utilization [44].

Plant water status was negatively affected by salt stress due to osmotic stress caused by excessive salt (NaCl) in the root zone [45]. The salt-sensitive rice (KDML105) was reported to suffer a significant reduction in leaf RWC at seedling [34] and vegetative [18] stages. However, seven out of ten genotypes of mature plants in this study, including KDML105, efficiently maintained the water status of the flag leaves (Figure 10A; Appendix A). This could be due to the combined mechanisms of water saving by reduced transpiration (Figure 9C), lowering of cellular water potential through osmotic adjustment (Figure 10B,C) and adaptation of root anatomical features for water acquisition under stress [46]. Salt stress also leads to overaccumulation of ROS, which attack macromolecules, particularly lipid components (lipid peroxidation) of cellular membranes, leading to the loss of membrane integrity and leakage of electrolytes (EL), making EL and malondialdehyde (MDA, a final product of lipid peroxidation) useful indicators of oxidative stress [45,47]. Among the ten rice genotypes, MDA levels remained largely unchanged under salinity, except in variety HN, which showed a significant increase (Figure 11B; Appendix A). Consistent with MDA, membrane integrity was not notably affected in most genotypes. However, HN exhibited a substantial 78% increase in EL, indicating severe membrane destabilization and oxidative damage. A previous report stated that salt-sensitive KDML105 at the vegetative stage exhibited a dramatic increase in both MDA and EL when subjected to 150 mM NaCl for 10 days [18], but these parameters in KDML105 at the reproductive stage remained stable, as shown in Figure 11A,B. This indicates that the ROS defense mechanisms are strengthened in mature plants at the reproductive stage.

Rice salt tolerance is achieved through three major physiological mechanisms to minimize salt-induced damage: osmotic adjustment (OA), Na^+^ exclusion and maintenance of Na^+^/K^+^ homeostasis, and ROS detoxification [48,49]. In this study, all ten rice genotypes performed OA under salt stress, which indicated that OA is one of the vital mechanisms to protect cells from salt-induced osmotic stress and maintain cell turgor [11]. The levels of OA varied considerably, and genotypes could be separated into two groups: (1) the low OA group (1.598 to 2.002 MPa) comprising PK, RD73, HDN, HN and TSKC1–144 and (2) the high OA group (2.283 to 2.576 MPa) comprising KDML105, GMS, LLR395, LPT and KG (Figure 10C, Appendix A). Under salt stress, OA involves the synthesis and accumulation of organic osmolytes such as sugars, sugar alcohols and proline in the cytosol to balance the reduced osmotic potential in the vacuoles due to high concentration of Na^+^ [50,51]. Exclusion of toxic Na^+^ from entering the shoots is considered the most crucial mechanism underlying salt tolerance in rice and is highly correlated with plant growth [52]. Based on Na^+^ concentration in flag leaves, the ten genotypes could be divided into two groups: (1) high Na^+^ exclusion (low Na^+^ content) comprising PK, HDN, TSKC1–144, and RD73 (having Na^+^ concentration of 1.09, 1.68, 1.98 and 2.16, respectively), and (2) low Na^+^ exclusion (high Na^+^ concentration) including the remaining six genotypes (Figure 11C, Appendix A). Within the low OA group, PK, RD73 and TCKC1-144 had highly expressed *SKC1* gene coding for the HKT1;5 transporter protein functioning in excluding Na^+^ from root xylem, leading to relatively low concentration of Na^+^ in leaf cells [36]. These genotypes exhibited a relatively low % increase in sugar (Figure 12A; Appendix A), i.e., low OA activity (Figure 10C). Contrastingly, the salt-sensitive KDML105 exhibited significantly higher Na^+^ than PK, RD73 and TSKC1–144 (Figure 11C); therefore, it needed to accumulate higher sugar osmolytes (Figure 12A; Appendix A) and hence possessed higher OA activity (Figure 10C). Among the local landrace rice, HDN displayed similar performance to PK, i.e., relatively low Na^+^ content (Figure 11C) and low OA (Figure 10C). This could be the reason for the high tolerance of HDN at the seedling stage (Figure 2B). Noticeably, salt-stressed KDML105 and RD73, which are genetically closely related, had similar stomatal conductance and transpiration rates (Figure 9B,C), but KDML105 showed much greater osmotic adjustment ability (Figure 10C), which may lead to better maintenance of RWC (Figure 10A). Therefore, the ten genotypes displayed different patterns of the two most important adaptive response mechanisms, ion exclusion and osmotic adjustment, which resulted in differential growth responses.

Salt stress adversely affected rice growth, as indicated by a significant reduction in plant height and total DW of all ten genotypes (except PK) and a significant increase in senescent and dead leaves (Figure 13). This could be related to the severe reduction in net photosynthesis rate combined with a reduction in active green leaf area due to limited leaf growth and expansion and accelerated leaf senescence [53]. Salt stress imposed at the reproductive phase during the early booting stage until harvest resulted in a considerable reduction in grain weight per plant in all genotypes (Figure 13G). The negative impact of salt stress on grain production was associated with (1) the reduction in grain number due to the harmful effects of Na^+^ in retardation of spikelet growth and development, pollen sterility, and impairment of fertilization and (2) the reduction in grain size and weight due to reduced photosynthesis of flag leaves and impairment of carbohydrate partitioning to developing grains [15,53]. Based on the percentage reduction in grain yield per plant (Figure 13G, Appendix A), rice genotypes could be separated into three groups: (1) low reduction (53 to 60%)—TSKC1–144, GMS, PK and RD73; (2) moderate reduction (75 to 76%)—HDN and HN; and (3) high reduction (86 to 93%)—LLR395, KDML105, LPT and KG. Effects of salinity on plant growth can also be classified into three groups based on the reduction in total plant DW, i.e., (1) low biomass reduction (3 to 31%)—PK, GMS and HDN; (2) moderate biomass reduction (41–48%)—RD73, KDML105, TSKC1–144, HN and LLR395; and (3) high biomass reduction (61 and 64%)—LPT and KG (Appendix A).

Data integration of biomass and grain yield reduction and the two most crucial salt tolerance mechanisms (Na^+^ exclusion and OA) enabled the separation of the ten genotypes into four response types (Figure 15). The type [A], which comprises PK, RD73, TSKC1–144 and HDN was characterized by high-efficiency Na^+^ exclusion, resulting in low to medium biomass and grain yield reduction. On the contrary, type [D] genotypes (LLR, KDML105, KG and LPT), which possess a low capacity for Na^+^ exclusion but perform strongly in OA, suffered medium to high biomass reduction and high grain yield reduction. The strong growth reduction in the type [D] genotypes could be attributed to the toxic effects of high Na^+^ in the tissues as well as the shortage of carbon sources for growth due to the high energy cost for synthesizing organic solutes for OA [10,54]. The local variety GMS, a type [B] response, displayed similar mechanisms as type [D] but recorded low reduction in biomass and grain yield, despite its high leaf Na^+^. This genotype probably possesses the tissue tolerance mechanism by which excess Na^+^ ions are efficiently sequestered into vacuoles via the highly active Na^+^/H^+^ antiporters (NHXs) on the vacuolar membrane, which actively pump Na^+^ from cytosol into vacuoles in exchange for H^+^ [55]. Fauzia et al. [56] recently reported a japonica rice variety (SZK), which displayed a tissue tolerance mechanism associated with high NHX gene expression. It accumulated high Na^+^ in roots, leaf sheath and blade tissues like salt-sensitive rice but maintained growth and dry weight comparable to salt-tolerant rice. The variety HN displayed type [C] response, i.e., low-efficiency Na^+^ exclusion as well as low OA, but still showed a moderate level of growth reduction. Tolerance mechanisms of this response type remain to be elucidated in the future.

In addition to deciphering differential salt tolerance mechanisms of rice genotypes at the reproductive stage, this research also revealed that salt tolerance at the seedling stage does not always perfectly reflect the performance at the reproductive stage. Some genotypes such as PK and RD73 were tolerant at both the seedling and reproductive stages. On the other hand, KG and LPT, which were tolerant at the seedling stage, were strongly impacted by salt stress at the reproductive stage. Among the ten genotypes, most physiological and agronomic parameters of KG and LPT were most highly disturbed. In contrast, TSKC1–144, which was sensitive at the seedling stage, became more tolerant at the reproductive stage and exhibited the least reduction in grain yield. This information is useful for farmers to optimize suitable agronomic management in saline fields for the optimum growth of certain rice genotypes at the seedling and reproductive stages. Among the 16 local Thai rice varieties tested in this study, GMS and HDN, with their good agronomic performance, are the most promising for further exploration of their field performance and could be a potential genetic resource for breeding salt-tolerance rice. The TSKC1–144 line is an interesting breeding material to be further developed into a salt-tolerant cultivar and to be employed as a potential parent for pyramiding genes for additional qualities. The officially released cultivar RD73, proven to possess cooking and eating qualities similar to KDML105 [57], is recommended for cultivation in moderately to highly saline environments in the northeast of Thailand, where KDML105 cannot thrive.

## 4. Materials and Methods

### 4.1. Study Site and Plant Materials

A total of 24 rice genotypes with contrasting levels of salt tolerance were collected from various research stations across Thailand. These included sixteen local Thai varieties, three improved cultivars/line, two widely grown commercial cultivars (KDML105 and PPT1), one KKU germplasm (LLR395), one standard salt-susceptible cultivar (IR29), and one standard salt-tolerant variety (Pokkali) (Table 1). KDML105 is the world-famous aromatic rice known to be salt-susceptible. RD73 is a genetically improved salt-tolerant cultivar developed by introgression of the *Saltol* QTL from Pokkali into the KDML105 genetic background, and was released by the Department of Rice, Thailand, in 2017 [57]. TSKC1–144 is a salt-tolerant backcross breeding line, derived from KDML105 introgressed with the *SKC1* gene from Pokkali [36]. The experiment was conducted in a greenhouse from November to June 2025 at the Department of Biology, Faculty of Science, Khon Kaen University (16.4743° N, 102.8293° E; 195 m above sea level). The maximum photosynthetically active radiation (PAR) of the natural light intensity was approximately 1300 µmol (photon) m^−2^ s^−1^, the midday temperature peaked at 34 °C, and the relative humidity (RH) ranged from 47% to 75%. The physicochemical properties of the soil are presented in Appendix A. Soil water status was adequately maintained to ensure well-watered conditions. Fertilization was applied at a rate based on soil analysis and nutrient requirements for rice [58].

### 4.2. Seedling Stage Experiment

Rice screening for salt tolerance level at the seeding stage, seeds of each rice genotype were sterilized with 2% sodium hypochlorite for 10 min and washed repeatedly with distilled water. All rice seeds were germinated in a plastic container (8 row × 6 columns) containing 50 g of loam soil per cell. Two seeds were sown per hole, and each genotype was planted in two holes within each container. The experimental design was a split plot in a randomized complete block design (RCBD) with five replications (n = 5) (Appendix A). The main plot of the experiment comprised different salt management treatments, while the sub-plot included rice genotypes (24 genotypes). For rice screening, the salt treatments were initiated by irrigating each container with 3 L of a 150 mM NaCl solution (EC water ≈ 15 dS m^−1^) at 14 days after germination (DAG). The salinity level was maintained through daily additions of pure water, monitored using a conductivity meter (ID1010, INDEX, Pullman, WA, USA). Rice plants were scored for salt stress damage after 7 and 10 days based on the protocol of the International Rice Research Institute [25,59]. The standard evaluation system (SES) scores of rice were given the scores from 1 to 9 based on growth, overall plant status, and leaf symptoms (white leaf tip, leaf browning and death) under salt stress, where a lower score (1) indicates tolerance and a higher score (9) denotes sensitivity (Appendix A). Growth parameters, including leaf greenness (SPAD), plant height, and total biomass, were determined at 24 DAG. All data were analyzed and classified rice varieties into 6 groups: high tolerance (HT), tolerance (T), moderate tolerance (MT), moderate susceptible (MS), susceptible (S), and highly susceptible (HS).

### 4.3. Reproductive Stage Experiment

To investigate the salt-tolerance physiological responses, mechanisms and growth at the reproductive stage, ten selected genotypes, including Go Main Surin (GMS), Hom Dang Nouy (HDN), Hom Noun (HN), LLR395, KDML105, Khao Gaew (KG), Leuang Puang Tawng (LPT), Pokkali (PK), RD73 and TSKC1–144, were germinated in a 70 × 70 cm plastic container containing 20 kg of loam soil. After 10 days, healthy rice seedings were transferred to plastic pots (25 cm height × 35 cm diameter), each filled with 5 kg of loam soil and containing two plants per pot. Salt treatments were initiated from the early booting to flowering stage by irrigating each pot with 2 L of a 150 mM NaCl solution (EC ≈ 15 dS m^−1^), with the salinity level maintained through daily additions of pure water for 19 days. During the grain-filling stage, the salt treatments were reduced from 15 to 12 dS m^−1^ and maintained at this level using pure water until harvest (28 days). The experimental design was a split plot in RCBD with eight replications. Data were collected on SPAD, effective quantum yield of PSII photochemistry (ΦPSII), electron transport rate (ETR), and leaf gas exchange in the destructive plants (replications #2, #4, #6, and #8; n = 4) at the flowering stage, following 14 days of salinity stress. After photosynthetic measurements, additional physiological parameters were determined, including chlorophyll content (Chl a, Chl b, and Chl a/b), relative water content (RWC), osmotic potential (OP), osmotic adjustment (OA), electrolyte leakage (EL), malondialdehyde (MDA), total sugar, and starch content. Growth parameters, including plant height, leaf, stem, root, grain, and total dry weight (DW), were measured in the non-destructive plants (replications #1, #3, #5, and #7; n = 4) at final harvest.

#### Data Collection for the Reproductive Stage Experiment

Photosynthetic pigments

SPAD was determined on two main-stem flag leaves in each replication (n = 4) using a chlorophyll meter (Minolta SPAD-502 Plus, Konica Minolta Inc., Osaka, Japan). For chlorophyll content, flag leaves from two plants of each replication (n = 4) were collected. Briefly, the flag leaf sample (0.1 g fresh weight) was homogenized in 5 mL of 80% acetone. The absorbance of the filtered solutions was measured at 645 and 663 nm (Hanon, Model i3, China). Chlorophyll a, b, and total chlorophyll were expressed as mg g^−1^ tissue fresh weight and calculated using the equations following Arnon [60] and Lichtenthaler [61]. Chl a, b, and total Chl values were calculated according to the following equations: Chl a = (12.7 (A663) − 2.69 (A645)) × (V/(1000 × W)), Chl b = ((22.9 (A645) − 4.68 (A663)) × (V/(1000 × W)), and Total Chl = ((20.2 (A645) + 8.02 (A663)) × (V/(1000 × W)), where V is the total volume of the filtered solution (mL), W is fresh weight of leaf tissue (g), and A645 and A663 are the absorbance values at 645 and 663 nm, respectively.

2.Chlorophyll fluorescence and leaf gas exchange

Photosynthetic performance, including light-response curves of Φ PSII, ETR and Pn, was measured on the middle part of the main-stem flag leaf of ten rice genotypes after 14 days of salt stress, with one plant per replication (n = 4). Chlorophyll fluorescence and leaf gas exchange of the control and stress plants were measured on sunny days using an infrared gas analyzer (IRGA) model Li-Cor 6400xt with an LED light source using a standard 2 × 3 cm leaf chamber (Li-Cor Inc., Lincoln, NE, USA). The measurement conditions were controlled every 120 s as follows: light intensity at different PAR levels of 1800, 1200, 600, 200, 50, and 0 μmol (photon) m^−2^ s^−1^, CO_2_ concentration at 400 μmol mol^−1^, temperature at 30 ± 2 °C, and RH at 60–65%. The flow rate was kept at 500 mL min^−1^.

3.Relative water content (RWC), osmotic potential (OP), and osmotic adjustment (OA)

The relative water content (RWC) was determined from the middle part of the flag leaf blade by cutting a 2 cm segment, immediately placed in a pre-weighed micro-tube, sealed and placed in a box. The tube was weighed to obtain the fresh weight (FW). The leaf segment was then floated on deionized water in a Petri dish placed under fluorescent light for 4 h at room temperature. Then, the leaf sample was re-weighed to obtain the turgid weight (TW) and subsequently dried in an oven at 80 °C for 48 h to determine the dry weight (DW). The relative water content (RWC) was calculated following Barrs and Weatherley [62] using the equation RWC (%) = (FW-DW)/(TW-DW) × 100.

To determine the osmotic potential (OP), the same leaves as those for RWC were used [11]. The main-stem flag leaves were immediately frozen in liquid nitrogen. The collected leaf samples were stored at −80 °C for further analysis of osmolality. Leaf sap was extracted from the thawed sample, and 10 µL of the leaf sap was used to determine the osmolality (c) using an osmometer (Vapro 5520, Wescor, South Logan, UT, USA). OP was calculated according to the Van’t Hoff equation: OP = −RTc, where OP is osmotic potential of the leaf sap (MPa), R is the gas constant (0.008341 L MPa mol^−1^ K^−1^), T is 298 K (room temperature at 25 °C) and c is osmolality of leaf sap (mmol/kg H_2_O).

Osmotic adjustment (OA) was calculated as the difference between the osmotic potential at full turgor (osmotic potential at 100% relative water content, OP_100_) of the control and stressed leaves. To calculate OP_100_, the equation of Turner et al. [63] was used assuming apoplastic water content of rice to be 18%, i.e., OP_100_ = [OP (RWC-18)]/82, where OP is the osmotic potential of the leaf sample and RWC is the relative water content of the leaf sample.

4.Electrolyte Leakage (EL) and Malondialdehyde (MDA)

The degrees of cell membrane injury induced by salinity stress were evaluated by determining the electrolyte leakage (EL) according to the method of Bajji et al. [64]. Five leaf segments (ca. 1 cm long) were put into 10 mL of deionized water in a test tube at room temperature for 24 h in the dark. The electrical conductivity (EC) of the water containing the leaked electrolytes was then measured to obtain the EC1 value using an EC meter (PL-700PCS GONDO, Taiwan, China). The tube containing leaf segments was then boiled for 15 min to damage the membrane and release all cellular electrolytes. After the tubes cooled to room temperature, the EC2 was measured. Electrolyte leakage (%) was calculated using the equation EL = (EC1/EC2) × 100.

Malondialdehyde (MDA) as an indicator of lipid peroxidation in rice leaves was determined following the method of Velikova et al. [65]. A 0.1 g fresh weight leaf sample was homogenized in 2.5 mL 0.1% (*w*/*v*) trichloroacetic acid (TCA; Sigma-Aldrich, WI, USA) solution. The homogenate solution (1 mL) was centrifuged at 10,000× *g* for 20 min. Supernatant 0.5 mL was added to 5 mL of 0.5% (*w*/*v*) thiobarbituric acid (TBA; Sigma-Aldrich, Milwaukee, WI, USA) in 20% TCA. The mixture was incubated in boiling water for 10 min, and the reaction was stopped by placing the reaction tubes in an ice bath for 5 min. Then, the solution was centrifuged at 10,000× *g* for 5 min. The absorbance of the supernatant was measured at 532 nm and 600 nm. The MDA content was calculated using the extinction coefficient of the red MDA-TBA product of 155 mM^−1^ cm^−1^. The MDA concentration in mM was calculated using the formula MDA = (A532 − A600)/155.

5.Total sugar and starch content

Soluble sugar was estimated using the Anthrone reagent following Luo and Huang [66], with slight modifications. A 0.1 g fresh flag leaf sample was weighed in a centrifuge tube, to which 1 mL of 80% ethanol was added. The sample was heated in an 80 °C water bath for 30 min, then centrifuged (12,000 rpm) for 10 min. The supernatant was collected, and the extraction was repeated twice for 10 min each. The supernatant was collected, and 80% ethanol was added to a total volume of 1 mL. Then, 10 µL of the extract diluted with 90 µL distilled water was mixed with 600 µL of freshly prepared Anthrone reagent (0.5 mM Anthrone (Thermo Fisher, Waltham, MA, USA) in 70% sulfuric acid), and the tubes were boiled for 12 min. The reaction was terminated by quick cooling on ice. The absorbance was measured at 620 nm. The total soluble sugar (mg g^−1^ FW) was quantified using glucose as a standard.

Starch content was determined by a modification of the method of Luo and Huang [66]. The residual materials that remained after the ethanol extraction of sugar were rinsed with 500 µL distilled water in a 1.5 mL microtube and 650 µL of cold 52% perchloric acid was added. The reactions were mixed by vortexing and then left standing at room temperature for 30 min. The samples were centrifuged at 13,000 rpm for 10 min, and the supernatants were collected. The supernatant was diluted with 100 µL distilled water, mixed with 600 µL of freshly prepared Anthrone reagent, and boiled for 12 min. The reaction was terminated by quick cooling on ice. The absorbance was measured at 620 nm. The sugars (mg g^−1^ FW) were quantified using glucose as a standard, and starch content was calculated using stock glucose 1 mg mL^−1^.

6.Growth, biomass and ion content

Plant height was measured at final harvest (plant age 110 days) from two plants in each replication (n = 4). For biomass measurement, two plants of each rice genotype (n = 4) were harvested 110 days after germination. For each plant, leaves, stems, roots and grain were separated, dried at 80 °C for 48 h (BF 720, INDER GmbH (Headquarters), Tuttlingen, Germany), or until weight was constant and then weighed to obtain dry weight. Leaves were separated into two parts: (1) green leaves and (2) combined senescent (yellowing) and dead leaves. At final harvest, sodium (Na^+^) and potassium (K^+^) content of dried leaf samples (0.1 g) was determined using atomic absorption spectroscopy (Corning, Model GBC932AAA, Slough, UK) after digesting in 10 mL of nitric acid at 300 °C followed by 5 mL perchloric acid at 200 °C, and 20 mL of 6 M hydrochloric acid.

### 4.4. Data and Statistical Analysis

The analysis of variance according to the spilt plot in RCBD was performed to assess the significance of quantitative changes in various parameters in the different rice genotypes and salinity managements. The main plot of the experiment was salinity management (control and salt stress), while the sub-plot was rice genotypes. For comparing rice genotypes in both saline conditions, the results were subjected Tukey’s honestly significant difference (HSD) for multiple comparisons of means at an alpha level of 0.05. All the graphs were taken using Sigmaplot Version 11.0 software (San Jose, CA, USA). The correlation among photosynthetic pigments (SPAD, Chl a, Chl b and Chl a/b), chlorophyll fluorescence parameters (Φ PSII and ETR), leaf gas exchange parameters (Pn, gs, Tr, WUEi, WUE, Ci and Ci/Ca), RWC, OP, OA, EL, MDA, Na^+^, K^+^ content, Na^+^/K^+^ and growth (i.e., plant height, dry weight of total leaves, green leaves, stems, roots and grains) of rice genotypes were conducted in each salinity management. Principal component analysis (PCA) and hierarchical cluster analysis (HCA) with a heatmap were used to group rice genotypes growing under different salinity treatments based on the mean physiological responses and growth data of each rice genotype. Pearson’s correlation, PCA and HCA were conducted using R version 3.4.3 and Rstudio version 2023.12.1.402 [67,68].

## 5. Conclusions

Twenty-four rice genotypes, including Thai local landrace, improved, and commercial cultivars, were evaluated for salt tolerance at the seedling stage. Ten selected genotypes (eight tolerant to moderately tolerant and two sensitive) were investigated for physiological and growth responses to high salinity during the early booting stage to harvest. While chlorophyll content remained relatively stable, salinity stress markedly impaired CO_2_ assimilation. All genotypes (except for one local variety, HN) were quite resistant to oxidative damage, resulting in non-significant increases in MDA content and EL. All genotypes displayed varying degrees of osmotic adjustment. Tolerant genotypes (PK, RD73, TSKC1–144 and HDN) displayed an efficient Na^+^ ion exclusion mechanism, resulting in low Na^+^ content. Low Na^+^ and Na^+^/K^+^ in PK, RD73 and TSKC1–144, regulated by salt tolerance genes located on the *Saltol* QTL, were associated with relatively low percentage reduction in most growth traits and grain yield. On the other hand, the elite commercial rice KDML105, lacking functional *Saltol* QTL, suffered a high grain yield reduction of 89%. The local Thai variety GMS showed good potential for cultivation in saline soils or as a gene donor, based on a similar percentage reduction in grain yield (58%) compared with PK (60%), RD73 (60%) and TSKC1–144 (53%). These results provide valuable insights for breeding programs aimed at developing high-yielding, salt-tolerant rice varieties suitable for saline-prone regions, especially in Northeast Thailand. This study also demonstrated that salt tolerance levels vary with growth stages, highlighting the importance of phenotyping at both growth stages during breeding. Moreover, for cultivars with different levels of salt tolerance at different growth stages, specific growing areas and suitable cultural management should be considered for maximum productivity. Future research should involve validating potentially salt-tolerant local varieties such as GMS and HDN under field conditions or incorporating the results into marker-assisted breeding.

## Figures and Tables

**Figure 1 plants-14-03748-f001:**
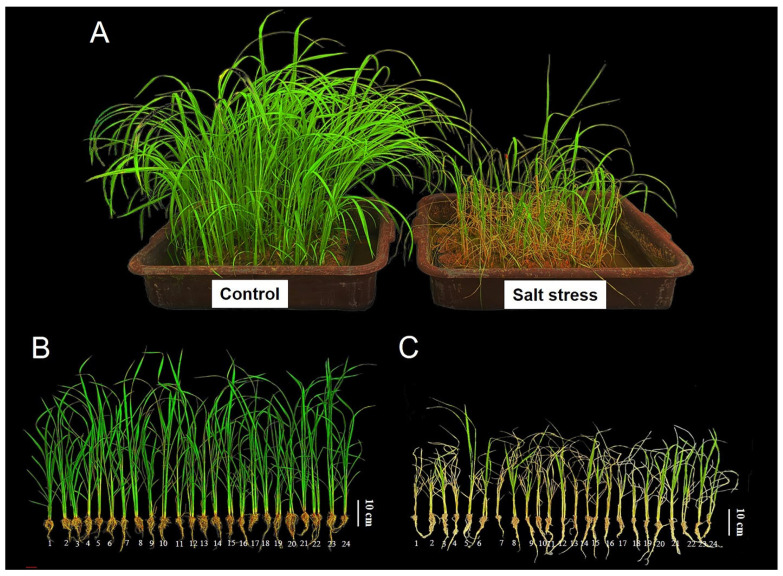
Rice plants at the seedling stage (24-day old) grown under non-saline and saline conditions (**A**). Twenty-four rice genotypes under non-saline (control, (**B**)) and saline conditions (**C**) displayed growth and salinity damage in each rice genotype under salt stress: (1) HMLN, (2) KDML105, (3) LLR395, (4) Ma Horn, (5) Pokkali, (6) Lou Tang, (7) Leuang Tah Yang, (8) Gon Gaew, (9) RD61, (10) Khao Supan, (11) IR29, (12) Leuang Puang Tawng, (13) Mali, (14) Hom Noun, (15) Pahn Tawng59, (16) Go Main Surin, (17) TSKC1–144, (18) PTT1, (19) Luang Pratahn, (20) RD73, (21) Puang Tawng, (22) Khao Gaew, (23) Pahn Tawng60, and (24) Hom Dang Nouy.

**Figure 2 plants-14-03748-f002:**
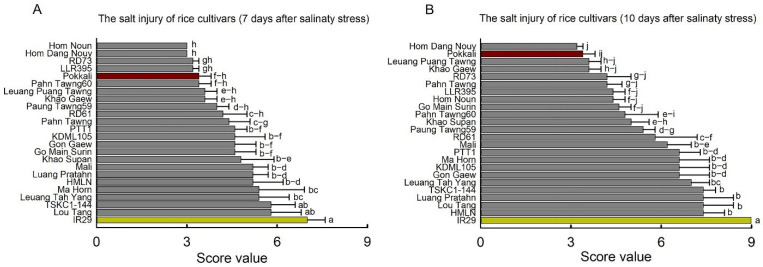
The salt injury scores of 24 rice genotypes grown under saline soil conditions (irrigation with 150 mM NaCl) after 7 days (**A**) and 10 days (**B**). The degrees of salt damage (SES scores) included 1—highly tolerant, 3—tolerant, 5—moderately tolerant, 7—susceptible, and 9—highly susceptible. Different lowercase letters indicate significant differences (*p* < 0.05 and *p* < 0.01) among the rice genotypes and between salinity treatments. The red and yellow columns represent the standard tolerance (Pokkali) and highly susceptible rice (IR29), respectively. Data are mean ± SE (n = 5).

**Figure 3 plants-14-03748-f003:**
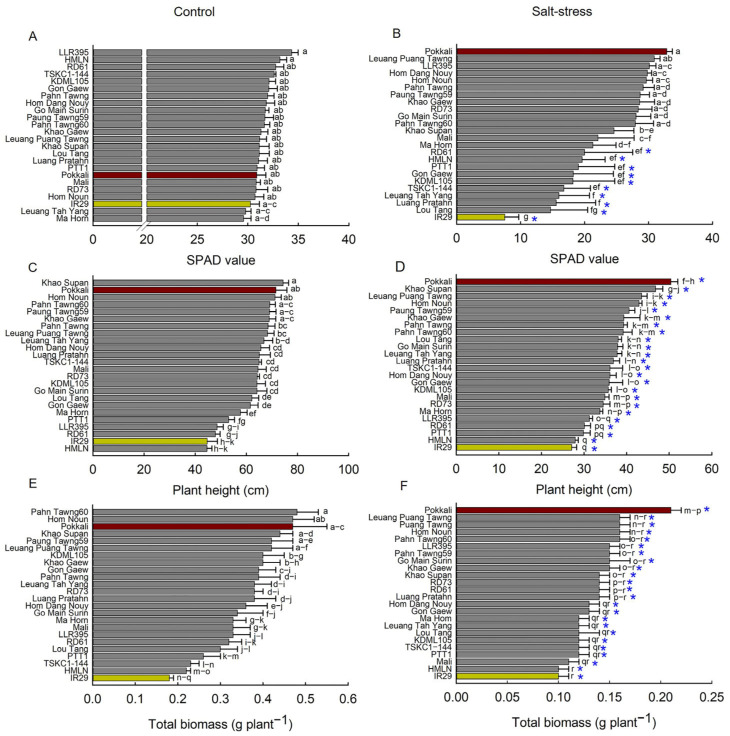
SPAD, plant height and total biomass of 24 rice genotypes grown under non-saline (**A**,**C**,**E**) and saline conditions (**B**,**D**,**F**). The significant difference (*p* < 0.05 and *p* < 0.01) among the rice genotypes under both salinity conditions are denoted with different lowercase letters. The red and yellow columns represent the standard tolerance (Pokkali) and highly susceptible rice (IR29), respectively. Significant differences between the control and salt stress treatments are denoted by *. Data are mean ± SE (n = 5).

**Figure 4 plants-14-03748-f004:**
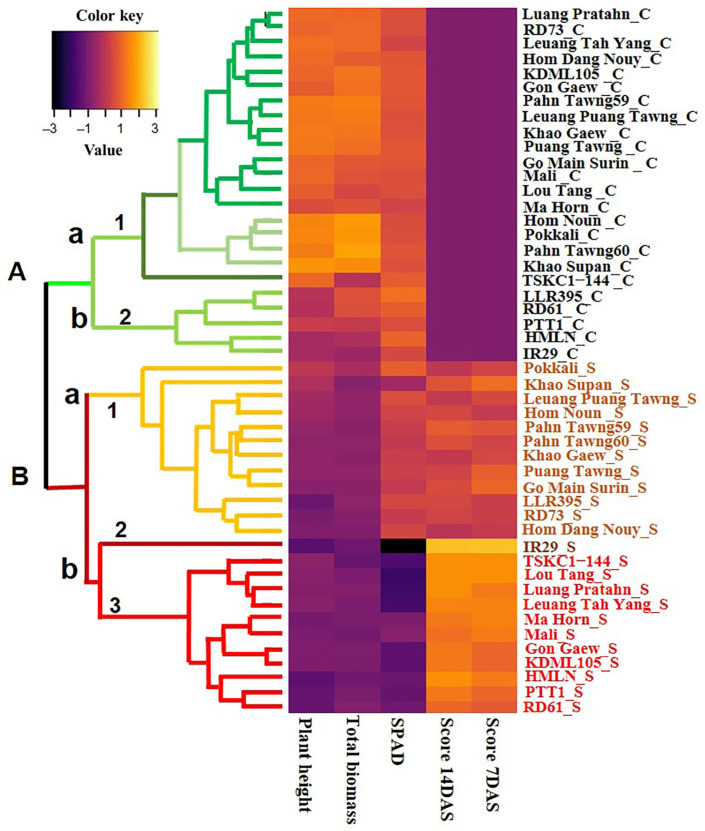
Hierarchical clustering analysis (HCA) explaining the responses of 24 local rice genotypes, including Go Main Surin, Gon Gaew, HMLN, Hom Dang Nouy, Hom Noun, IR29, LLR395, KDML105, Khao Gaew, Khao Supan, Lou Tang, Leuang Puang Tawng, Luang Pratahn, Leuang Tah Yang, Ma Hom, Mali, Puang Tawng, Pahn Tawng60, Pahn Tawng59, Pokkali, PTT1, RD61, RD73 and TSKC1–144 growing under control and saline conditions. The HCA columns correspond to dependent variables, whereas the rows correspond to different conditions (rice genotypes growing under control (C) and saline (S) conditions).

**Figure 5 plants-14-03748-f005:**
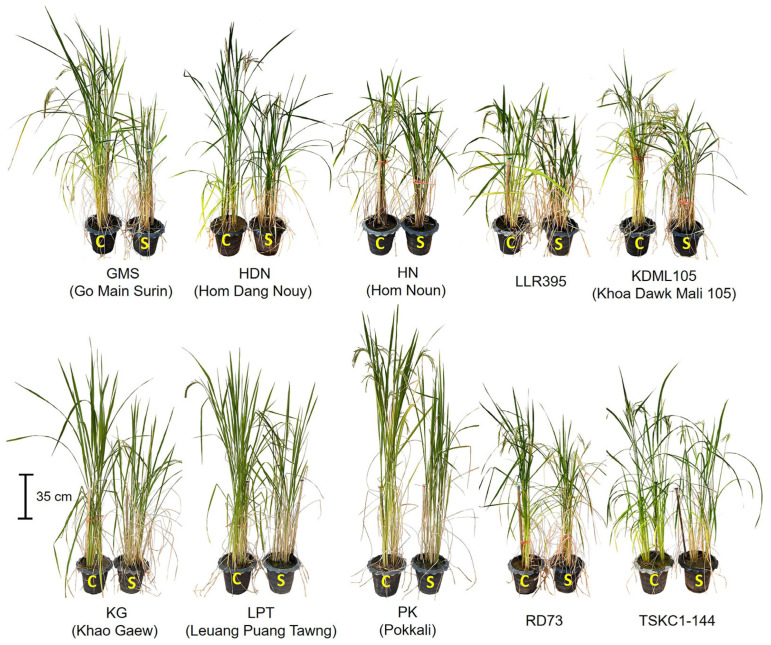
Rice plants at the reproductive stage (82-day old) grown under non-saline and saline conditions. After 19 days of salinity treatments, 10 rice genotypes under non-saline (control, C) and saline conditions (S) displayed plant height and salinity damage under salt stress: Go Main Surin (GMS), Hom Dang Nouy (HDN), Hom Noun (HN), LLR395, KDML105, Khao Gaew (KG), Leuang Puang Tawng (LPT), Pokkali (PK), RD73, TSKC1–144.

**Figure 6 plants-14-03748-f006:**
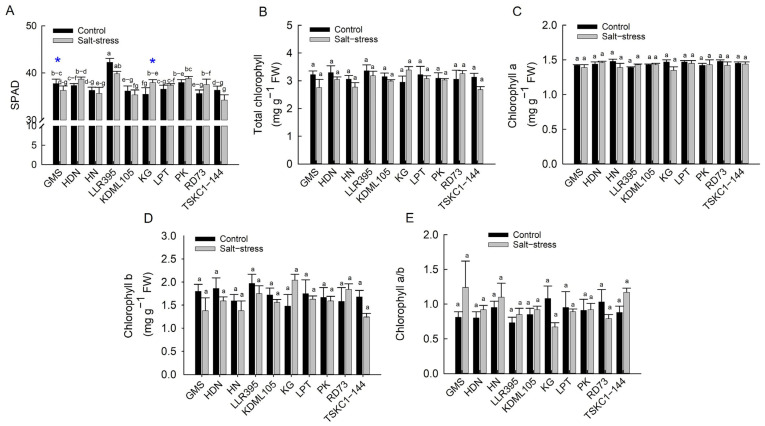
Photosynthetic pigments in flag leaves, including SPAD index (**A**), total chlorophyll (**B**), chlorophyll a (**C**), chlorophyll b (**D**) and chlorophyll a/b (**E**) of 10 rice genotypes. Plants were grown under control conditions and salt stress (irrigated with 150 mM NaCl instead of water for 19 days during the reproductive stage). Different lowercase letters indicate significant differences (*p* < 0.05 and *p* < 0.01) among rice genotypes and between salinity treatments. Significant differences between the control and salt stress treatments are denoted by *. Data are mean ± SE (n = 4).

**Figure 7 plants-14-03748-f007:**
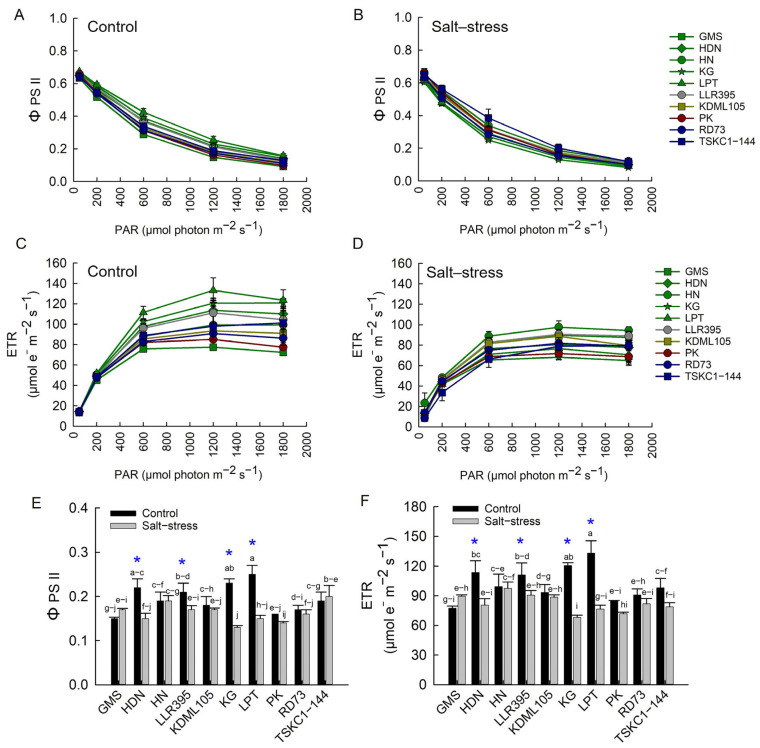
Effective quantum yield of PSII photochemistry (Φ PSII) and electron transport rate (ETR) at varying light intensity (**A**–**D**) and under illumination with a light intensity of 1200 µmol (photon) m^−2^ s^−1^ (**E**,**F**) in flag leaves of 10 rice genotypes at the flowering growth stage. The Φ PSII/I, ETR/I curves, Φ PSII and ETR were measured in rice plants growing under the control (non-saline) and salt stress (irrigated with 150 mM NaCl instead of water for 14 days) conditions. Different lowercase letters indicate significant differences (*p* < 0.05 and *p* < 0.01) among rice genotypes and between salinity treatments. Significant differences between the control and salt stress treatments are denoted by *. Data are mean ± SE (n = 4).

**Figure 8 plants-14-03748-f008:**
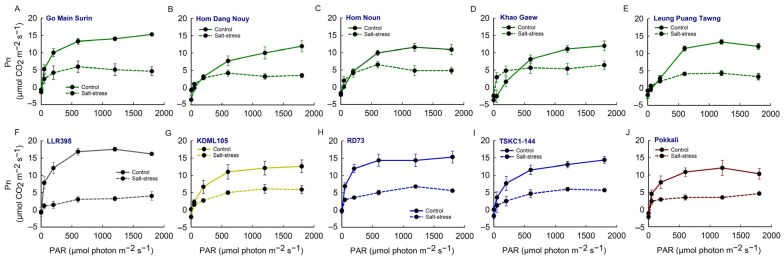
Photosynthetic light-response curves (Pn/I curve) in flag leaves of ten rice genotypes, including Go Main Surin (GMS, (**A**)), Hom Dang Nouy (HDN, (**B**)), Hom Noun (HN, (**C**)), Khao Gaew (KG, (**D**)), Leuang Puang Tawng (LPT, (**E**)), LLR395 (**F**), KDML105 (**G**), RD73 (**H**), TSKC1–144 (**I**) and Pokkali (PK, (**J**)). The Pn/I curves were measured in rice plants growing under control (non-saline) and salt stress (irrigated with 150 mM NaCl instead of water for 14 days during the reproductive growth stage) conditions. The green, gray, yellow, red and blue lines represent local Thai rice varieties, KKU germplasm, commercial rice, the standard salt-tolerant check, and improved salt-tolerant rice, respectively. Data are mean ± SE (n = 4).

**Figure 9 plants-14-03748-f009:**
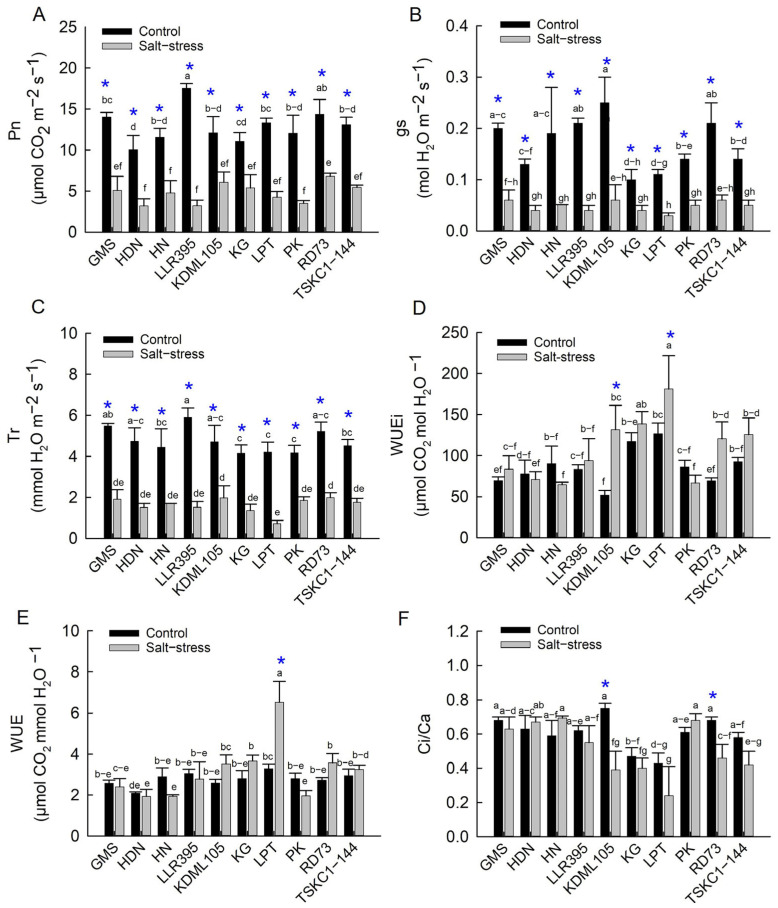
Leaf gas exchange, including net photosynthesis rate (Pn, (**A**)), stomatal conductance (gs, (**B**)), transpiration rate (Tr, (**C**)), intrinsic water use efficiency (WUEi, (**D**)), water use efficiency (WUE, (**E**)), and the intercellular to ambient CO_2_ ratio (Ci/Ca, (**F**)) investigated under PAR of 1200 µmol (photon) m^−2^ s^−1^ in flag leaves of ten rice genotypes. Plants were grown under control conditions and salt stress (irrigated with 150 mM NaCl instead of water for 14 days during the reproductive stage). Different lowercase letters indicate significant differences (*p* < 0.05 and *p* < 0.01) among rice genotypes and between salinity treatments. Significant differences between the control and salt stress treatments are denoted by *. Data are mean ± SE (n = 4).

**Figure 10 plants-14-03748-f010:**
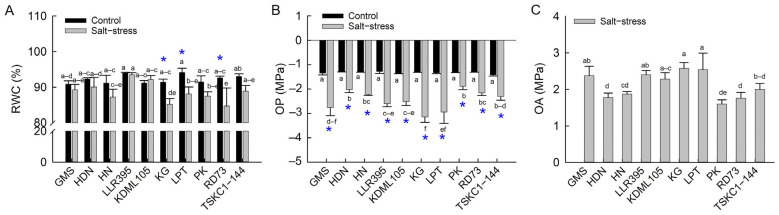
Relative water content (RWC, (**A**)), osmotic potential (OP, (**B**)) and osmotic adjustment (OA, (**C**)) in flag leaves of ten rice genotypes. Plants were grown under control conditions and salt stress (irrigated with 150 mM NaCl instead of water for 19 days during the reproductive stage). Different lowercase letters indicate significant differences (*p* < 0.05 and *p* < 0.01) among rice genotypes and between salinity treatments. Significant differences between the control and salt stress treatments are denoted by *. Data are mean ± SE (n = 4).

**Figure 11 plants-14-03748-f011:**
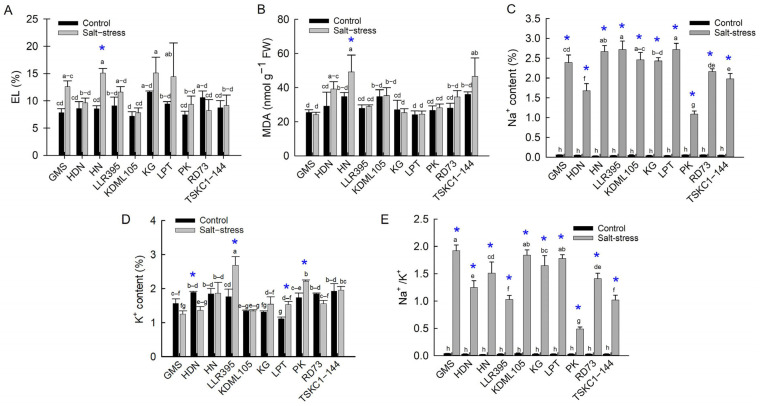
Electrolyte leakage (EL, (**A**)), malondialdehyde (MDA, (**B**)), sodium ion content (**C**), potassium ion content (**D**) and the ratio of sodium to potassium ions (**E**) in the flag leaves of ten rice genotypes. Plants were grown under control conditions and salt stress (irrigated with 150 mM NaCl instead of water for 19 days during the reproductive stage). Different lowercase letters indicate significant differences (*p* < 0.05 and *p* < 0.01) among rice genotypes and between salinity treatments. Significant differences between the control and salt stress treatments are denoted by *. Data are mean ± SE (n = 4).

**Figure 12 plants-14-03748-f012:**
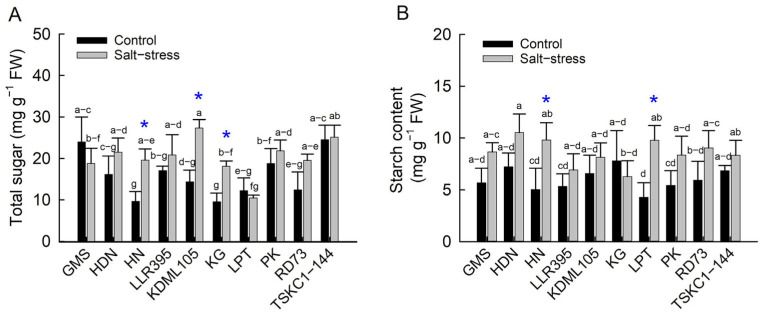
Total sugar (**A**) and starch (**B**) in the flag leaves of ten rice genotypes. Plants were grown under control conditions and salt stress (irrigated with 150 mM NaCl instead of water for 19 days during the reproductive stage). Different lowercase letters indicate significant differences (*p* < 0.05 and *p* < 0.01) among rice genotypes and between salinity treatments. Significant differences between the control and salt stress treatments are denoted by *. Data are mean ± SE (n = 4).

**Figure 13 plants-14-03748-f013:**
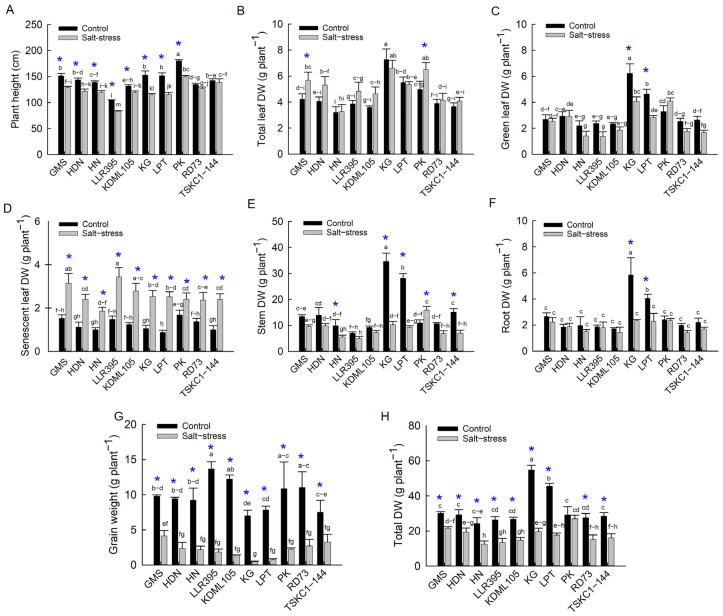
Growth parameters, including plant height (**A**) and dry weight (DW) of total leaf (total leaf DW, (**B**)), green leaf (GL DW, (**C**)), senescent leaf (SL DW, (**D**)), stem DW (**E**), root DW (**F**), gain (**G**), and total DW (**H**) of ten rice genotypes at the final harvest (110 DAG). Plants were grown under control conditions and salt stress. Different lowercase letters indicate significant differences (*p* < 0.05 and *p* < 0.01) among rice genotypes and between salinity treatments. Significant differences between the control and salt stress treatments are denoted by *. Data are mean ± SE (n = 4).

**Figure 14 plants-14-03748-f014:**
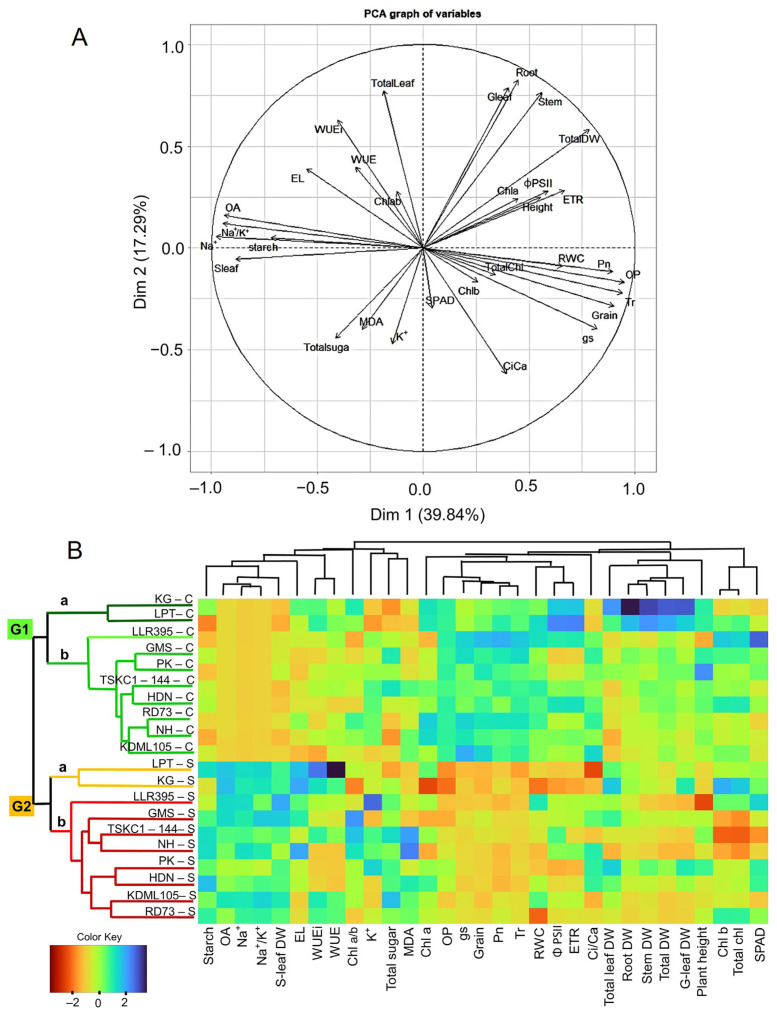
Principal component analysis (PCA, (**A**)) and hierarchical clustering analysis (HCA, (**B**)) explaining the responses of ten rice genotypes, including GMS, HDN, HN, LLR395, KDML105, KG, LPT, PK, RD73, and TSKC1–144, growing under non-saline and saline conditions. The PCA and HCA indicate the variations based on the mean values of SPAD index, total chlorophyll (chl), chl a, chl b, chl a/b, effective quantum yield of PSII photochemistry (Φ PSII), electron transport rate (ETR), net photosynthesis rate (Pn, stomatal conductance (gs), transpiration rate (Tr), intrinsic water use efficiency (WUEi), water use efficiency (WUE), and the intercellular to ambient CO_2_ ratio (Ci/Ca) at a light intensity of 1200 µmol (photon) m^−1^ s^−1^, relative water content (RWC), osmotic potential (OP), osmotic adjustment (OA), electrolyte leakage (EL), malondialdehyde (MDA), total sugar, starch content, plant height and dry weight (WD) of total leaves, green leaves (Gleaf), senescent leaf (Sleaf), stem, root, grain, and total DW. The columns of HCA correspond to dependent variables, whereas the rows correspond to different treatments (genotypes under different salt treatments). Low numerical values are yellow, while high numerical values are blue (see the scale at the right corner of the heatmap).

**Figure 15 plants-14-03748-f015:**
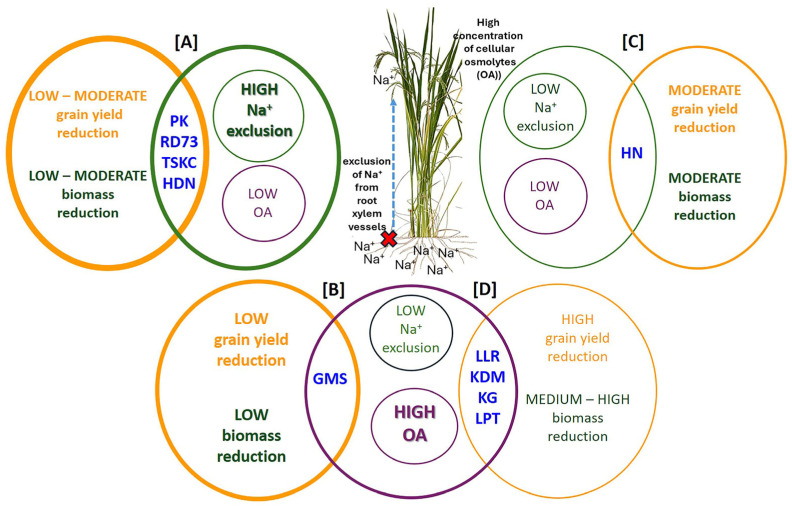
Diagrammatic representation of the differentiation of ten rice genotypes into four groups (A, B, C and D) based on the levels of the two most crucial adaptive salt tolerance mechanisms, Na+ exclusion and osmotic adjustment (OA), and the degrees of reduction in growth and grain yield when rice plants were subjected to salt stress from the early booting stage to harvest. [PK, Pokkali; TSK, TSKC1–144; HDN, Hom Dang Nouy; GMS, Gomain Surin; HN, Hom Noun; LLR, LLR395; KDM, Khao Dawk Mali 105 (KDML105); KG, Khao Gaew; LPT, Leuang Puang Tawng].

**Table 1 plants-14-03748-t001:** Information on the 24 rice genotypes used in these experiments.

Rice Name	GS.No./ Code	Information	Institutes
Go Main Surin (GMS)	LLR377	local Thai variety	Agronomy, KKU
Gon Gaew (GG)	1698	local Thai variety	Rice Department of Thailand
HMLN	–	local Thai variety	Rice Department of Thailand
Hom Dang Nouy (HDN)	LLR054	local Thai variety	Agronomy, KKU
Hom Noun (HN)	LLR055	local Thai variety	Agronomy, KKU
IR29	–	standard susceptible check	Agronomy, KKU
LLR395	LLR395	KKU germplasm	Agronomy, KKU
Khao Dawk Mali 105 (KDML105)	13744	commercial cultivar	Rice Department of Thailand
Khao Gaew (KG)	6152	local Thai variety	Rice Department of Thailand
Khao Supan (KS)	2296	local Thai variety	Rice Department of Thailand
Leuang Puang Tawng (LPT)	7214	local Thai variety	Rice Department of Thailand
Leuang Tah Yang (LTY)	14685	local Thai variety	Rice Department of Thailand
Lou Tang (LT)	LLR143	local Thai variety	Agronomy, KKU
Luang Pratahn (LP)	2975	local Thai variety	Rice Department of Thailand
Ma Hom (MH)	LLR309	local Thai variety	Agronomy, KKU
Mali	LLR210	local Thai variety	Agronomy, KKU
Pahn Tawng59 (PT59)	7559	local Thai variety	Rice Department of Thailand
Pahn Tawng60 (PT60)	7560	local Thai variety	Rice Department of Thailand
Pokkali (PK)	–	salt-tolerant standard check	Rice Science Center
PTT1	–	commercial cultivar	Agronomy, KKU
Puang Tawng (PT)	18442	local Thai variety	Rice Department of Thailand
RD61	–	improved cultivar	Agronomy, KKU
RD73	24867	improved salt-tolerant cultivar	Rice Department of Thailand
TSKC1–144	144	improved salt-tolerant line	Biology, KKU

Note: Agronomy, KKU = Department of Agronomy, Faculty of Agriculture, Khon Kaen University, Khon Kaen; Rice Department of Thailand = National Rice Germplasm Operation and Storage Center, Pathum Thani Rice Research Center; Biology, KKU = Salt-tolerant Rice Research Group, Department of Biology, Faculty of Science, Khon Kaen University; Rice Science Center = Rice Science Center, National Center for Genetic Engineering and Biotechnology (BIOTEC), Thailand.

## Data Availability

The original contributions presented in this study are included in the article/Appendix A. Further inquiries can be directed to the corresponding author.

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
