# Peer review of "Physiological and Growth Responses of Thai Rice Genotypes to High Salinity Stress at the Seedling and Reproductive Stages"

_plants, 2025, doi:10.3390/plants14243748_

Round 1
Reviewer 1 Report
Comments and Suggestions for Authors
This paper systematically investigates the salt tolerance responses of Thai rice germplasm at the seedling and reproductive stages, and the topic holds significant theoretical and practical value. The experimental design is rigorous, and the data are substantial, providing rich support particularly at the physiological mechanism level. The study successfully reveals the diversity of salt tolerance strategies among different genotypes, offering new resources and insights for salt-tolerant breeding. The main issues include: a lengthy introduction, lack of key methodological information, simplistic repetition of data in the results analysis, and insufficient connection between the discussion and results.
Introduction
The content is overly lengthy, and the relevance of some background information to the core research objectives needs to be strengthened.
The logical flow between paragraphs is not smooth. It is recommended to streamline literature content that is less relevant to the manuscript, focusing on the issues of rice salt stress, the diversity of salt tolerance mechanisms, and the value of local germplasm resources.
Strengthen the logical connections between paragraphs, considering a progressive structure from "research background – existing gaps – objectives of this study."
Materials and Methods
1. The method description in lines 940–942 is extremely vague. "Watering" does not specify the irrigation method, which may lead to issues of salt accumulation or dilution; the control treatment method is not described; and it is not mentioned whether there was a pre-adaptation process. It is recommended to clearly specify the details of the salt treatment: irrigation volume, frequency, method of maintaining salt concentration, control group treatment (nutrient solution or pure water?), and whether a gradually increasing concentration stress adaptation scheme was used, providing as much detail as possible.
2. The salt treatment description in lines 959–962 has similar but more severe issues. Continuous irrigation with saline water for 19 days may lead to intense salt accumulation in the soil; reducing the irrigation water concentration does not necessarily mean the soil solution concentration decreases simultaneously. It is recommended to specify the reproductive stage salt treatment plan in detail: irrigation volume per application, frequency, whether soil EC values were monitored to confirm stress levels, and the physiological basis for reducing the concentration.
3. The units of salt solution used in the seedling and reproductive stages are inconsistent and should be unified throughout the text.
4. The actual EC values of the pot soil were not measured to confirm that the stress level reached the target. Methods and results of EC monitoring should be supplemented to verify the stress intensity.
5. The dimension description in line 956, "plastic tray (70 x 70 cm)," does not match the context. It is recommended to verify and correct it to "plastic pot" or "plastic container."
6. Photosynthesis measurements and growth assessments were conducted on different replicates—was this random or selective? It is recommended to clearly state whether this destructive sampling design involved random allocation of replicates.
7. The OP100 calculation formula assumes an apoplastic water content of 18%, without citing the basis. It is recommended to provide the literature source for this assumption to enhance persuasiveness.
8. The input data types for PCA and HCA analyses were not specified. It is recommended to clearly indicate whether raw data, standardized data, or relative values were used.
9. Detailed parameters for chlorophyll fluorescence measurements need to be supplemented with complete descriptions of all parameters.
Results
1. The presentation of results is primarily a listing of data, lacking refinement of core findings. It is recommended to begin each subsection with 1–2 sentences summarizing the core findings, followed by data support.
2. Most of the results analysis only describes "what," lacking explanation of "what this means."
3. The comparison of key genotypes is not effectively highlighted, making it difficult for readers to quickly grasp the different salt tolerance strategies. It is recommended to clearly establish a genotype comparison framework and maintain consistent comparisons throughout the text.
4. The various results are relatively isolated, with no active connection made between physiological indicators, ion data, and growth/yield. It is recommended to use connecting words and transitional sentences.
Discussion Section
1. The content is repetitive, with discussions of the same genotype scattered across different subsections, weakening the argument. It is recommended to thoroughly reorganize the discussion structure, remove redundant parts, and highlight key findings. For example, the point about "the trade-off between ion exclusion and osmotic adjustment" is not sufficiently refined—this core argument could be strengthened, clearly explaining that "efficient ion exclusion is key to salt tolerance at the reproductive stage."
2. Integration of data is insufficient, and conclusions from different figures and tables are not effectively connected. It is recommended to integrate all key findings and use a comprehensive schematic diagram to holistically present the paper’s conclusions.
3. Some conclusions are expressed too absolutely, such as asserting that certain parameters "cannot be used for genotype differentiation."
Conclusion Section
1. The depth of discussion is insufficient, merely summarizing facts without refining theoretical contributions.
2. The expression is not rigorous, such as stating "low OA genotypes are all salt-tolerant," which overlooks exceptional cases.
There are non-academic expressions such as "dramatically" and "interesting to note." It is recommended to conduct a comprehensive language polish, using objective and precise academic expressions.
Author Response
Comments and Suggestions for Authors
This paper systematically investigates the salt tolerance responses of Thai rice germplasm at the seedling and reproductive stages, and the topic holds significant theoretical and practical value. The experimental design is rigorous, and the data are substantial, providing rich support particularly at the physiological mechanism level. The study successfully reveals the diversity of salt tolerance strategies among different genotypes, offering new resources and insights for salt-tolerant breeding. The main issues include: a lengthy introduction, lack of key methodological information, simplistic repetition of data in the results analysis, and insufficient connection between the discussion and results.
Introduction
The content is overly lengthy, and the relevance of some background information to the core research objectives needs to be strengthened.
The logical flow between paragraphs is not smooth. It is recommended to streamline literature content that is less relevant to the manuscript, focusing on the issues of rice salt stress, the diversity of salt tolerance mechanisms, and the value of local germplasm resources.
Strengthen the logical connections between paragraphs, considering a progressive structure from "research background – existing gaps – objectives of this study."
Author response: Thank you so much for your constructive comments. I totally agree and the INTRODUCTION has been revised so that the overall length was reduced by approximately 30%. It is now composed of Paragraph 1 - the importance of rice, problems of salinity particularly in the northeast Thailand. Paragraph 2 – Effects of salt stress on rice and the mechanisms to cope with salt stress. Paragraph 3 – Research gap – lack of information of physiological responses and QTL identification at reproductive stage. Paragraph 4 – Objectives.
Materials and Methods
1. The method description in lines 940–942 is extremely vague. "Watering" does not specify the irrigation method, which may lead to issues of salt accumulation or dilution; the control treatment method is not described; and it is not mentioned whether there was a pre-adaptation process. It is recommended to clearly specify the details of the salt treatment: irrigation volume, frequency, method of maintaining salt concentration, control group treatment (nutrient solution or pure water?), and whether a gradually increasing concentration stress adaptation scheme was used, providing as much detail as possible.
Author response: Thank you very much for bringing up this important point. In the revised manuscript, ‘Watering’ has been replaced with ‘Irrigation,’ and we have provided a more detailed description of the salt treatment protocol. Line 765-798
- The salt treatment description in lines 959–962 has similar but more severe issues. Continuous irrigation with saline water for 19 days may lead to intense salt accumulation in the soil; reducing the irrigation water concentration does not necessarily mean the soil solution concentration decreases simultaneously. It is recommended to specify the reproductive stage salt treatment plan in detail: irrigation volume per application, frequency, whether soil EC values were monitored to confirm stress levels, and the physiological basis for reducing the concentration.
Author response: Thank you for this valuable comment. We appreciate your concern regarding potential salt accumulation during continuous irrigation and the need for a clearer description.
We have revised the manuscript to provide a detailed of the salt treatment protocol. Specifically, we have now included: Irrigation volume per application, irrigation frequency and duration of the treatment and water electrical conductivity (EC) monitoring procedures. Line 813-820
- The units of salt solution used in the seedling and reproductive stages are inconsistent and should be unified throughout the text.
Author response: Thank you for pointing this out. We have carefully revised the manuscript to ensure that the units of salt solution used in both the seedling and reproductive stages are now consistent throughout the text.
- The actual EC values of the pot soil were not measured to confirm that the stress level reached the target. Methods and results of EC monitoring should be supplemented to verify the stress intensity.
Author response: Thank you for highlighting this important point. The actual EC values of the soil at the pre-planting and the EC of the irrigation water at seedling stage, early booting to flowering stage, and milky stage to final harvest have been explained in Supplementary Table S12.
- The dimension description in line 956, "plastic tray (70 x 70 cm)," does not match the context. It is recommended to verify and correct it to "plastic pot" or "plastic container."
Author response: Thank you for pointing out this inconsistency. The term “plastic tray (70 × 70 cm)” has now been corrected to “plastic container” to accurately reflect the materials used in the experiment. Line 792,812
- Photosynthesis measurements and growth assessments were conducted on different replicates—was this random or selective? It is recommended to clearly state whether this destructive sampling design involved random allocation of replicates.
Author response: Thank you for raising this point. In the revised manuscript, we now explicitly state which replicates were used for destructive and which for non-destructive growth measurements.
Line 812-813, 824-830
- The OP100 calculation formula assumes an apoplastic water content of 18%, without citing the basis. It is recommended to provide the literature source for this assumption to enhance persuasiveness.
Author response: Thank you for highlighting this issue. In the revised manuscript, we have now added the appropriate literature citation “Turner N. C., O'Toole J. C., Cruz R. T., Yambao E. B., Ahmad S., Namuco O. S., et al. (1986). Response of seven diverse rice cultivars to water deficit. II. Osmotic adjustment, leaf elasticity, leaf extension, leaf death, stomata conductance and photosynthesis. Field Crop Res. 13, 273–286.” Line 880
- The input data types for PCA and HCA analyses were not specified. It is recommended to clearly indicate whether raw data, standardized data, or relative values were used.
Author response: Thank you for this helpful comment. In the revised manuscript, we added the input data type ‘mean values’ to the Results and Methods sections for the PCA and HCA analyses. Line 554, 950
- Detailed parameters for chlorophyll fluorescence measurements need to be supplemented with complete descriptions of all parameters.
Author response: Thank you for pointing this out. Detailed parameters for chlorophyll fluorescence measurements were added. Line 286-287
Results
1. The presentation of results is primarily a listing of data, lacking refinement of core findings. It is recommended to begin each subsection with 1–2 sentences summarizing the core findings, followed by data support.
Author response: We agree that the Results section would benefit from clearer emphasis on the core findings. In the revised manuscript, each subsection now begins with 1–2 sentences summarizing the main results, highlighting rice genotypes significance.
Most of the results analysis only describes "what" lacking explanation of "what this means."
Author response: Thank you for pointing this out. We tried to add the meaning in each result section wherever possible.
The comparison of key genotypes is not effectively highlighted, making it difficult for readers to quickly grasp the different salt tolerance strategies. It is recommended to clearly establish a genotype comparison framework and maintain consistent comparisons throughout the text.
Author response: Thank you for this important suggestion. In the revised version, we have summarized the main results for each section and clearly highlighted the significant responses of rice genotypes under salt treatments by denoting them with * in the figures.
The various results are relatively isolated, with no active connection made between physiological indicators, ion data, and growth/yield. It is recommended to use connecting words and transitional sentences.
Author response: The connection between physiological parameters has been added. For example:
Discussion Section
The content is repetitive, with discussions of the same genotype scattered across different subsections, weakening the argument. It is recommended to thoroughly reorganize the discussion structure, remove redundant parts, and highlight key findings. For example, the point about "the trade-off between ion exclusion and osmotic adjustment" is not sufficiently refined—this core argument could be strengthened, clearly explaining that "efficient ion exclusion is key to salt tolerance at the reproductive stage."
Author response: The discussion part has been reduced in detail and reorganized. We also tried to reduce the redundancy as much as possible.
Integration of data is insufficient, and conclusions from different figures and tables are not effectively connected. It is recommended to integrate all key findings and use a comprehensive schematic diagram to holistically present the paper’s conclusions.
Author response: Important/ significant data have been integrated and highlighted in the form of diagram (Figure 15) which summarized differential mechanisms of genotypes.
Some conclusions are expressed too absolutely, such as asserting that certain parameters "cannot be used for genotype differentiation."
Author response: "cannot be used for genotype differentiation." has been removed.
Conclusion Section
1. The depth of discussion is insufficient, merely summarizing facts without refining theoretical contributions.
Author response: Discussion has been improved to integrate important mechanisms with growth and yield responses.
The expression is not rigorous, such as stating "low OA genotypes are all salt-tolerant," which overlooks exceptional cases.
Author response: The discussion on OA has been improved for clearer understanding.
Comments on the Quality of English Language
There are non-academic expressions such as "dramatically" and "interesting to note." It is recommended to conduct a comprehensive language polish, using objective and precise academic expressions.
Author response: Those expressions have been removed.
Reviewer 2 Report
Comments and Suggestions for Authors
Overall comments
The MS entitled “Physiological and Growth Responses of Thai Rice Genotypes to High Salinity Stress at Seedling and Reproductive Stages” seems to well well-written. In my opinion, the MS should be accepted for publication after addressing a very important issue.
Specific comments
In this experiment, many diverse rice genotypes were used. The reproductive stage of the diverse rice genotypes might be different and obviously there might be significant differences at the flowering stage of the genotypes. So, imposition of salt stress at the early booting stage, particularly at the same date, should not represent the specific desirable growth phase (reproductive stage) for all the rice genotypes. Therefore, the results obtained through this phenotyping technique might not reflect the actual response and results of the variable genotypes reflecting reproductive stage.
Although precise phenotyping techniques at the reproductive stage for variable genotypes is quite difficult, however, method was established by Ahmadizadeh et al. (2016: https://link.springer.com/article/10.1007/s40502-016-0268-6).
Author Response
Comments and Suggestions for Authors
Overall comments
The MS entitled “Physiological and Growth Responses of Thai Rice Genotypes to High Salinity Stress at Seedling and Reproductive Stages” seems to well well-written. In my opinion, the MS should be accepted for publication after addressing a very important issue.
Specific comments
In this experiment, many diverse rice genotypes were used. The reproductive stage of the diverse rice genotypes might be different and obviously there might be significant differences at the flowering stage of the genotypes. So, imposition of salt stress at the early booting stage, particularly at the same date, should not represent the specific desirable growth phase (reproductive stage) for all the rice genotypes. Therefore, the results obtained through this phenotyping technique might not reflect the actual response and results of the variable genotypes reflecting reproductive stage.
Author response: Thank you for highlighting this important point. The ten genotypes we selected were photosensitive and reached the panicle initiation stage within the time frame of 1 to 3 days different, and salt solution was introduced at early booting stage when the young panicles were approximately 1-2 cm.
Although precise phenotyping techniques at the reproductive stage for variable genotypes is quite difficult, however, method was established by Ahmadizadeh et al. (2016: https://link.springer.com/article/10.1007/s40502-016-0268-6).
Author response: Thank you so much for introducing this important publication for us. We have now cited this paper (number 15). This is extremely useful for our experimental planning in the future.
Reviewer 3 Report
Comments and Suggestions for Authors
In order to select the target Thai rice which are tolerant at seedling and reproductive growth stage, twenty-four rice genotypes were analyzed for physiological and growth responses to high salinity stress at seedling and reproductive stages. The research is meaningful in rice production. However, there were several questions as follows,
- The stress response analysis should be conducted. The response=(trait value under stress condition)/(trait value under control condition). Because that if A line has much high biomass under control condition than B line, the stress induced the A decreased much more percentage than B, even A has a higher value than B under stress condition, we don’t think A line has higher stress tolerance than B.
- Please discuss the reason for the different salt tolerance performance of two lines which introgressed with Saltol QTL and SKC1, respectively.
- For reproductive stage experiment, the the flowering stage were same or not among ten genotypes? Since the growth environments affect the Pn greatly.
Author Response
Comments and Suggestions for Authors
In order to select the target Thai rice which are tolerant at seedling and reproductive growth stage, twenty-four rice genotypes were analyzed for physiological and growth responses to high salinity stress at seedling and reproductive stages. The research is meaningful in rice production. However, there were several questions as follows,
- The stress response analysis should be conducted. The response =(trait value under stress condition)/(trait value under control condition). Because that if A line has much high biomass under control condition than B line, the stress induced the A decreased much more percentage than B, even A has a higher value than B under stress condition, we don’t think A line has higher stress tolerance than B.
Author response: Thank you very much for bringing up this important point. For most parameters, different genotypes under control had different values, so we used percentage changes under stress compared with the control when comparing responses among genotypes. The detailed percentage changes in all parameters can be found in Supplementary Tables. Moreover, in all graphs we added * (asterisks) wherever the difference in means between stress and control was significantly different.
- Please discuss the reason for the different salt tolerance performance of two lines which introgressed with Saltol QTL and SKC1, respectively.
Author response: Both RD73 and TSKC1-144 had improved ion homeostasis (lower Na, lower Na/K) as the results of introgressed Saltol QTL or SKC1 (gene located on Saltol). The reason they perform differently could be because RD73 received the larger piece of DNA of Saltol locus which contain very large number of genes, some could pose negative effects on SKC1 gene function.
- For reproductive stage experiment, the flowering stage were same or not among ten genotypes? Since the growth environments affect the Pn greatly.
Author response: Thank you very much for highlighting this important point. The ten genotypes we selected were photosensitive and reached the panicle initiation stage within the time frame of 1 to 3 days different, and salt solution was introduced at early booting stage when the young panicles were approximately 1-2 cm.
Reviewer 4 Report
Comments and Suggestions for Authors
1.The manuscript presents extensive physiological data across rice growth stages. Notably, for several indicators (e.g., EL, MDA, Na+/K+), resistant varieties do not consistently show higher values than sensitive ones. This raises key questions: is physiological resistance governed by a single factor or multiple interacting factors? Furthermore, can any specific physiological indicator reliably serve as a standalone marker for assessing resistance levels?
2.The manuscript observes that some tolerant genotypes with low OA activity (e.g., PK, RD73) exhibited efficient Na+ exclusion. However, this conclusion would be strengthened by supporting references on the general relationship between OA levels and resistance mechanisms.
3.In Figure 13, it is noteworthy that some varieties show higher values under salt stress than in the control. The reliability of these specific data points should be re-examined.
4.The conclusion mentions "high salinity (12–15 dS m⁻¹)". To ensure consistency, this specific salinity range should be standardized and used throughout the manuscript.
5.The clarity of the manuscript could be enhanced by making the language more concise, particularly in the Results and Discussion sections.
Comments on the Quality of English LanguageThe clarity of the manuscript could be enhanced by making the language more concise, particularly in the Results and Discussion sections.
Author Response
Comments and Suggestions for Authors
- The manuscript presents extensive physiological data across rice growth stages. Notably, for several indicators (e.g., EL, MDA, Na+/K+), resistant varieties do not consistently show higher values than sensitive ones. This raises key questions: is physiological resistance governed by a single factor or multiple interacting factors? Furthermore, can any specific physiological indicator reliably serve as a standalone marker for assessing resistance levels?
Author response: Rice is generally sensitive at the germination and early vegetative stage, but becomes more tolerant during reproductive stage (that is why some parameters for example, chlorophyll and lipid peroxidation did not change upon salt stress where previous experiments on vegetative or seedling stage showed salt damage in these parameters). The ten genotypes we selected on the basis of tolerance level at seedling stage based on standard salt injury scores, eight were tolerant – moderately tolerant, only two (KDML105 and TSKC1-144) were sensitive. At reproductive stage both KDML105 and TSKC1-144 became more tolerant so the tolerant varieties did not show consistently higher values than KDML105. The results showed that only two parameters, Na+ ion concentration (and Na/K) and osmotic adjustment, significantly changed in all ten genotypes and showed differential responses among genotypes and can be used as reliable physiological indicators.
- The manuscript observes that some tolerant genotypes with low OA activity (e.g., PK, RD73) exhibited efficient Na+ exclusion. However, this conclusion would be strengthened by supporting references on the general relationship between OA levels and resistance mechanisms.
Author response: Thank you for highlighting this important point. In the revised manuscript, we have added relevant references (number 54). The relationship between Na concentration and OA was discussed in Lines 679 – 686.
- In Figure 13, it is noteworthy that some varieties show higher values under salt stress than in the control. The reliability of these specific data points should be re-examined.
Author response: We have re-examined the data and confirmed the data. The total biomass generally reduced under salt stress with the exception of Pokkali which is highly salt tolerant.
- The conclusion mentions "high salinity (12–15 dS m⁻¹)". To ensure consistency, this specific salinity range should be standardized and used throughout the manuscript.
Author response: Thank you for pointing this out. We have reviewed the manuscript and ensured that the salinity level is now reported consistently in the Abstract and Methods (section 4.3, Reproductive stage experiment). This standardization improves clarity and prevents any confusion regarding the applied stress level. Line
- The clarity of the manuscript could be enhanced by making the language more concise, particularly in the Results and Discussion sections.
Author response: Thank you so much for your concern. We tried to make the language more concise especially where the data were complex.
Comments on the Quality of English Language
The clarity of the manuscript could be enhanced by making the language more concise, particularly in the Results and Discussion sections.
Round 2
Reviewer 1 Report
Comments and Suggestions for Authors
The authors have provided comprehensive and high-quality responses to the reviewers' comments. The revised manuscript is scientifically rigorous and logically coherent, particularly in the Discussion section, where integrated analysis has led to the proposal of a new classification framework for salt tolerance types. While the Conclusions section could be further refined, it meets the necessary standards. I recommend acceptance for publication.
Reviewer 3 Report
Comments and Suggestions for Authors
Thank you for the reply, the research work is meaningful.